



# Impact of dust in PMIP-CMIP6 mid-Holocene simulations with the IPSL model

Pascale Braconnot[1], Samuel Albani[1,2], Yves Balkanski[1], Anne Cozic[1], Masa Kageyama[1], Adriana Sima[3], Olivier Marti[1], and Jean-Yves Peterschmitt[1]

*Correspondence to* : Pascale Braconnot (Pascale.braconnot@lsce.ipsl.fr)

[1]LSCE-IPSL, unité mixte CEA-CNRS-UVSQ, Université Paris-Saclay, Orme des Merisiers, bât. 714, 91191 Gif-sur-Yvette Cedex, France

[2] Department of Environmental and Earth Sciences, University of Milano-Bicocca, Milano, Italy

[3]LMD-IPSL, 4 place Jussieu, Université Paris-Sorbonne, 75 005 Paris Cedex 05

Abstract.

We investigate the impact of reduced dust during mid-Holocene using simulations with the IPSL model . We consider simulations where dust is either prescribed from an IPSL PI simulation or from CESM

simulations (Albani et al., 2015). In addition, we also consider an extreme mid Holocene case where dust is suppressed. We focus on the estimation of the dust radiative effects and the relative responses of the African and Indian monsoon, showing how local dust forcing or orography affect atmospheric temperature profiles, humidity and precipitation. Compared to previous simulations with the IPSL model the results show only minor improvements for the mid Holocene simulation over large regions despite the fact that the IPSLCM6-

LR version of the IPSL model is in better agreement with modern observations. The effect of dust has little impact on the model-data comparisons. Our analyses confirm the peculiar role of dust radiative effect over bright surfaces such as African deserts compared to other regions due to the change of sign of the dust radiative effect at the top-of-atmosphere for high surface albedo. We also highlight a strong dependence on the dust pattern. In particular the relative dust forcing between West Africa and the middle east impacts the

relative climate response between India and Africa and between Africa, the western tropical Atlantic and the



Atlantic meridional circulation. Dust patterns should thus receive a wider attention to fully understand the changes in the dust cycle and forcing during mid Holocene.

## 1   Introduction

The mid-Holocene climate is a long-standing focus in the Paleoclimate Modeling Intercomparisons project

(1995; Kageyama et al., 2018). This climatic period is characterized by increased vegetation and moisture in now dry or semi-arid regions in Africa and Asia (COHMAP-Members, 1988; Prentice and Webb, 1998). Increased vegetation and water bodies are recognized as the footprints of increased inland advection of monsoon flow and monsoon rainfall (Joussaume et al., 1999). The large-scale increase of boreal summer precipitation which is now well understood, results from the changes in seasonal insolation induced by the

multi-millenial variations of Earth's orbital parameter. But despite more than 20 years of active research the PMIP3-CMIP5 generation of climate models didn't properly represent the amount of precipitation changes in the Sahel-Sahara region (Harrison et al., 2015; Joussaume et al., 1999; Perez-Sanz et al., 2014). Several factors, involving model biases, vegetation, soil moisture or dust feedbacks are key candidates to explain the mismatch between the model results and the paleoclimate reconstructions and will be investigated in more

depth during the 4[th] phase of PMIP (Kageyama et al., 2018).
Until recently, prognostic simulations for the dust cycle during the mid-Holocene with global Earth System models and comparison with observations have been carried out only by a few modeling groups with different approaches, only some of them in comparison with observations, yielding very different results (Albani and Mahowald, 2019; Hopcroft and Valdes, 2019; Liu et al., 2018; Pausata et al., 2016; Thompson

et al., 2019). In general, the focus has been on the response of dust emissions in North Africa following variations in climate and surface conditions (e. g. Egerer et al., 2018). On the other hand, the data assimilation approach developed by Albani et al. (2015) is based on a global compilation of paleodust archives and simulations with the CESM fully-coupled model to derive global reconstructions of the dust cycle, with the intent of focusing on the evaluation of dust impacts on climate (Albani and Mahowald,

50   2019).



Despite the different points of view, the results raised several questions on the dust-climate interactions and the way the response depends on the land surface conditions, with evidence that vegetation or lake cover play a role not only on dust emissions, but also on the radiative effect of dust and thereby on the atmospheric circulation and monsoon precipitation (Egerer et al., 2018; Messori et al., 2019; Pausata et al., 2016). It also

raises questions on how the different representations of dust in climate models impacts the magnitude of the dust feedback on mid Holocene precipitation (Hopcroft and Valdes, 2019). An aspect still lacking full investigation is how the differences in dust patterns considering the global distribution of dust beyond Africa are involved in climate teleconnections far from dust source regions, arising from the magnitude of regional response or the dust transport and the global atmospheric distribution of dust.

The role of dust is receiving a larger attention as part of the 4[th] Phase of the Paleoclimate Modeling Intercomparison project (Kageyama et al., 2018). Only a few modeling groups will be able to run mid-Holocene simulation with fully interactive climate-dust models that account for dust loading and deposition as a function of weather, vegetation and soil humidity conditions, atmospheric dust transport and three-dimensional interaction with radiation and clouds. To allow for the  participation of a large number of

modelling groups, a simplified protocol has been proposed that accounts for the dust interaction with radiation and clouds while prescribing the three dimensional characteristics of atmospheric dust from previous mid-Holocene simulations with the NCAR model (Albani et al., 2015; Otto-Bliesner et al., 2017). Here, we test this protocol using the IPSL model with the aim to investigate:

-  How large is the mid Holocene dust forcing and how does it affect African and Indian monsoon
     -  How the dust used as reference in different models when running mid mid-Holocene simulation affect the results. We address this here by comparing simulations using Albani's reference dust (Otto-Bliesner et al., 2017), instead of the pre-industrial (PI) dust field simulated with the INCA-IPSL model (Lurton et al., 2020).

- Whether it is possible to identify a typical pattern associated with dust changes that could be evaluated using paleoclimate reconstruction of temperature and precipitation.

Dust fields and the mid-Holocene sensitivity experiments using the IPSL model are presented in section 2.



Section 3 discusses the relative effect of insolation and dust on the model large scale hemispheric and land-sea contrasts, as well as the impact of dust on the tropical rain belt and the Afro-Asian monsoon.

## 2  Model and experiments

### 2.1  The IPSL model and the reference PMIP4-CMIP6 mid-Holocene simulation

We use version IPSLCM6A-LR of the Earth System Model (ESM) developed at Institut Pierre Simon Laplace (IPSL) (Boucher et al., 2020). This is the reference version for the suite of CMIP6 simulations from IPSL (Eyring et al., 2016). This model is an Earth system model that couples the atmosphere, ocean, land-
surface and sea-ice components through the energetic and hydrological cycles, as well as through biogeochemical cycles. The model experimental set up for the mid-Holocene simulations is the same as for pre-industrial CMIP simulations (Lurton et al., 2020). The atmosphere and land surface are represented with a resolution of 2.5° in longitude, 1.125° in latitude and 79 vertical levels. The ocean has a resolution of 1° with a refined grid at the equator and at the North Pole and 75 vertical levels. In these simulations the carbon
cycle is interactive in all components, except for the atmospheric composition for which is prescribed. In this configuration, the carbon fluxes between the different compartments need to adjust to the prescribed atmospheric values.

Compared to the previous reference version of the model (IPSLCM5A-LR, (Dufresne et al., 2013), all model components and model resolution have been improved. The reader is referred to the different
manuscripts presenting the new model version, the way the different forcing factors were introduced in the model and the model performances for a more complete description (Boucher et al., 2020; Lurton et al., 2020).

In this model version, natural and anthropogenic aerosols are fully interactive with the radiative code. In order to have a computationally efficient version of the model, we retained only two natural components of
the aerosols, namely: dust and sea-salt.  The size distribution of these two components is treated using a modal scheme that comprises one coarse insoluble mode to represent dust following Schulz et al. (1998), and three soluble modes for sea-salt: accumulation, coarse and super-coarse as described in Guelle et al. (2001). The dust 3D atmospheric concentrations fields are prescribed with monthly resolution, based on





climatological values from simulations with the INCA model for present-day conditions (Lurton et al., 2020).

The initial state for the mid-Holocene simulation is the year 1850 (January 1[st]) of the CMIP6 reference preindustrial simulation with the same model version (Boucher et al., 2020). It corresponds to the same initial state as the one used for one of the members of the IPSL CMIP6 historical simulation. The Earth's orbital and the atmospheric composition were switched to the values provided by Otto-Bliesner et al. (2017) for the mid-Holocene (Table 1). This mid-Holocene simulation follows exactly the PMIP4-CMIP6 protocol for mid-Holocene. A small difference with the estimation of the forcing provided in Otto-Bliesner et al. (2017) is due to the fact that the pre-industrial control simulation does not have Earth's parameters values for 1850, but the ones for circa 1987 (Lurton et al., 2020). The solar constant is set to 1361.2 Wm$^{-2}$.

The model was then run for 350 years to adjust to the mid-Holocene boundary conditions, which is sufficient to bring the climate in equilibrium for mid-Holocene conditions (Fig. 1, red curve). However, we note that the carbon cycle is not fully equilibrated and manifests a small drift in the deep ocean (not shown). Most surface variables adjust in about 100 years. Note that the model produces large decadal variations (Fig. 1), which imposes to characterize the mean annual cycle using at least 100 years. We chose 150 years in the following.

The last state of this simulation is used as initial state for the reference PMIP4-CMIP6 mid-Holocene simulation. This reference was run for 550 years, the last 50 years having high frequency outputs for the analyses of extremes of to provide the boundary conditions for future regional simulations. This reference simulation is called CMIP6.PMIP.IPSL.IPSL-CM6A-LR.midHolocene.r1i1p1f1 in the ESGF database and r1i1p1f1 (f1), or MHREF in the text below (Table 1)

### 2.2 Pre-industrial and mid Holocene dust fields for simulations with the IPSL model

The most striking feature of the Holocene dust cycle is the drastic reduction of dust emissions from today's major dust source, North Africa, compared to late glacial and late Holocene and present levels (e.g. deMenocal et al., 2000; McGee et al., 2013; Palchan and Torfstein, 2019). Dust emissions were probably also lower in the middle East (e. g. Pourmand et al., 2004), while more scattered signals emerge from East Asia (e. g. Kohfeld and Harrison, 2003). Dust emissions in the Southern Hemisphere were higher than in the





late Holocene, but much lower compared to the glacial period (Lambert et al., 2008). These and many more
observational constraints were compiled at the global level, also considering the particle size range and
distributions, and served as the basis for "data assimilation" in combination with the CESM model (Albani
et al., 2015), to provide the PMIP4 reference field for the mid-Holocene (Otto-Bliesner et al., 2017).

Before being uplifted from source regions, dust particles consist of mineral aggregates of several hundred
microns of diameter. Through bombardment, mineral aggregates are fractionated into smaller particles
which diameters are generally between 1 and 50 um that can be transported over large distances of hundreds
to thousands of kilometers. Each aerosol model has its own representation of these mineral dust particles
Adapting the original "Albani/CESM" PMIP4 datasets (Otto-Bliesner et al., 2017) for dust to the IPSL

model requires thus adapting the size distribution from a sectional representation to a modal one. Hence, the
amount and horizontal spatial variability of the original "Albani/CESM" PMIP4 datasets are very similar to
the actual prescribed datasets used in this work, although they are not identical. Figure 2 provides a
comparison of the interpolated reference dust fields (Albani0k and Albani6k) to the observational dataset
used to constrain the original PMIP4 reference dust fields (Albani et al., 2015). In particular, the ratio in

model dust loads is compared to the ratio in dust deposition (<10 μm diameter) from the observation. It is
possible to see clearly the -/+/- pattern in dust loads over North Africa/West Asia/East Asia, associated to
the dust load changes between mid-Holocene and preindustrial. It highlights that most of the changes are in
agreement with observed changes, except in East Asia or north America where the degree of agreement is
heterogeneous (Fig. 2). Extensive evaluation of these original simulated dust fields against observations is

available in Albani et al. (2016; 2015).

To obtain these files, we first had to transfer the information on the four bins provided in the 0.1-10 μm
diameter range to the model size distribution of the target IPSL model (1 mode: MMD 2.5 μm and SD 2.0).
The dust mass associated to the size distribution in the original dataset was remapped to the IPSL modal
aerosol scheme, for each grid box in the 3D dust atmospheric concentration field. One implication of this

procedure is that, compared to CESM simulations, the IPSL simulations have less dynamical variability of
Aerosol Optical Depth (AOD) and Dust Radiative Effect (DRE) as a function of changes in size
distributions with size. This is caused by the homogenization needed for the dust interactions with radiation
in the IPSL model, due to the fixed optical properties of the single mode. In addition, the vertical



discretization of the original model consists of 26 vertical layers compared whereas they are 79 layers for

the target model. Therefore, the 3D dust concentration fields are interpolated in the vertical accordingly. Monthly fields of dust load and concentrations provided for PMIP4 and computed using the CAM model (Albani et al., 2015) were thus re-gridded from the resolution of CAM4 (horizontal 288x192 (1.25 deg. x 0.94 deg.) with 26 layers in the vertical) to the IPSLCM6-LR resolution of 144x143 (2.5 deg. x 1.28 deg.) and 79 vertical layers. When the mid-pressure of IPSLCM6 was delimited by 2 vertical levels from CAM4,

the fields were interpolated between these two levels. If a mid-layer from IPSLCM6-LR was below (resp. above) the lowest (resp. highest) mid-layer of CAM4, we assigned to this layer the values of dust fields from the lowest (resp. highest) CAM4 level.

The global dust load for the MHREF, Albani0k, and Albani6k simulations is 18.7, 14.6, and 12.8 Tg respectively. For comparison, the original dust loads for "Albani/CESM" pre-industrial and mid-Holocene

are 20 and 18 Tg, respectively (Albani and Mahowald, 2019; Otto-Bliesner et al., 2017). Interestingly, the size distributions and optical properties of these Albani and Mahowald (2019) dust files interpolated on the IPSL model size distribution and spatial resolution are similar to the reference version of the IPSL model. Therefore, the differences among the simulations from this work are related to the 3D spatial distribution of dust (Fig. 3).  In contrast, differences between simulations from this work using "Albani/CESM" dust and

the original in Albani and Mahowald (2019) , can be interpreted as due to differences in other features differing in the CESM and IPSL models, including dust-related (size distributions, optical properties, vertical distributions) and non-dust-related (radiative transfer, boundary layer, etc.) processes and parameterizations (Table 2).

**2.3   Dust sensitivity experiments**

We performed 3 sensitivity experiments (Table 1), using the same model set up and initial state as the reference simulation MHREF (f1). The first sensitivity experiment (referred to as f2 or NODUST below) allows us to study the extreme case in which no dust would be present at the mid-Holocene. This provides the largest change we can expect from the dust forcing. This simulation of the CMIP6 database is the mid-

Holocene member r1i1p1f2 (Table 1). In this simulation the dust load and the 3D dust distribution are set to



0. All other aerosols are kept as in the mid-Holocene reference simulation. We then examine the effect of having mid-Holocene dust instead of preindustrial dust in the mid Holocene simulations using Albani's dust fields presented in section 2.2 as proposed by Otto-Bliesner et al (2017). However, the 3D-distribution of Albani's preindustrial dust field is different from the IPSL one. In particular MHREF has higher dust loads

over North Africa, the Thar desert, and East Asia, whereas Albani0k and Albani6k have higher dust loads than MHREF in the middle-East and central Asia (Fig. 3). Therefore, the differences in the pre-industrial dust distribution need to be considered in the analyses because they also affect the Albani's mid Holocene dust distribution. We show below that indeed this effect is larger than the effect of mid-Holocene reduction of dust discussed by Albani and Mahowald (2019). We thus performed two simulations. The r1i1p1f3 (f3,

Albani0k) simulation tests the effect of replacing the IPSL dust fields by Albani's dust fields obtained for the preindustrial climate. The last simulation, r1i1p1f4 (f4, Albani6k), is run with Albani's dust fields obtained for the mid-Holocene, and tests the effect of mid-Holocene dust when compared to f3. For simplicity, we will use below the following nomenclature for the experiments: MHREF, NODUST, Albani0k and Albani6k, as this has the advantage to make a direct link with the corresponding dust files

(Table 1).

Since the dust forcing is small when globally averaged, these simulations equilibrate rapidly to the forcing induced by the replacement of a dust distribution by another and are quite stable (Fig. 1) They were run only for 300 years, except r1i1p1f4 that is 290 year-long (Table 1). Indeed, the globally averaged net (SW+LW) TOA Dust effective radiative perturbation is in the range -0.11 to -0.07 and W m$^{-2}$, with larger negative

radiative perturbation for pre-industrial dust as expected (Table 2). We call this measure effective perturbation (ERP) and not Effective Radiative Forcing (ERF) as in Forster et a. (2016) ,because it is estimated by comparing each simulation with a simulation in which dust is suppressed (i.e. NODUST here) over the first 50 year of the simulation. It thus includes dust forcing and fast adjustment feedbacks. These radiative perturbations are small but slightly higher than in the original CESM1 simulations for which it is -

0.06 and -0.03 W m$^{-2}$ for PI and MH, respectively (Albani and Mahowald, 2019). The comparison of the sum of absolute values of dust ERP at each grid point provides higher values by avoiding compensations between regions of negative and positive Dust ERP. It ranges between |0.56| and |0.80| W m$^{-2}$ in the IPSL model against |0.27| and |0.23| W m$^{-2}$ in CESM, indicating stronger geographical gradients in the response



(Table 2). This comparison confirm that the interpolation of Albani et al. (2015) dust file on the IPSL model
leads to a slight reduction of dust load and dust effective radiative forcing, but that the difference in radiative
forcing is larger between two different models due to differences in the representation of dust and of their
interaction with atmospheric radiation.

The mean annual cycle discussed for these estimates and below refers to averages over 150 years. To avoid
any artificial differences due to possible long-term adjustment, the mean seasonal cycle for the reference
simulation r1i1p1f1, is computed over the same period. It corresponds to the period between year 2250 and
2400 in Figure 1 (i.e. year 50 to 200 of the mid Holocene simulations stored on ESG with dates 185001-
204912 in model calendar years. In the following we use the longer simulations to estimate the uncertainties
resulting from final model adjustment and variability. However, during this common period all simulation
have a similar long-term variability (Fig. 1).

### 3    Model adjustment and characteristics of the mid-Holocene climate

### 3.1    Adjustment to the mid-Holocene orbital and trace gases forcing

The dust experiments are designed to test the impact of dust on the mid-Holocene climate. We thus first
discuss the differences between the mid-Holocene and the preindustrial. The mid-Holocene changes in
obliquity and precession have an impact on the spatio-temporal (seasonal) distribution of insolation, but not
on its global annual mean. The rapid adjustments seen on most variable plotted in Figure 1 between year
1850 and 1950 is thus primarily due to the model adjustment to the reduced trace gases (Brierley et al.,
2020; Otto-Bliesner et al., 2017). It corresponds to a reduction (about 0.4 °C) in 2m annual mean
temperature (Fig. 1b). The reduced temperature induced by changes in insolation and trace gases is
amplified by a slight increase in land albedo, and an increase sea-ice thickness in the Southern Hemisphere
(not shown). It is also compensated by a slight reduction in surface evaporation (Fig. 1c), which contributes
to reduce the total precipitation amount (Fig. 1d). The reduced $CO_2$ and temperature lead to a global
reduction of the total biomass (not shown) despite an increased carbon flux out of the atmosphere to biomass
productivity (Fig. 1g).



These annual mean changes are however small compared to the large seasonal variations induced by precession that enhance seasonality in the northern hemisphere and reduce it in the southern hemisphere. The model reproduces the well-known patterns of the simulated mid-Holocene climate (Fig. 4 and Fig. 5)(e. g. Braconnot et al., 2007). The reduced December January February (DJF) insolation induces a general cooling over land, maximum in the subtropics and at the sea-ice margin. It also induces a tropical ocean

cooling (Fig. 4 a. and b.). These effects cause a strengthening of the Northern hemisphere winter monsoon, leading to a drying over land and to wetter conditions over the ocean, in particular in the Indian ocean (Fig.4 a and b). They also produce a slight southward shift of the ITCZ in the Pacific and in the Atlantic (Fig. 5 a. and b.). During mid Holocene late spring, the increased solar radiation in the northern hemisphere, when the tropical ocean is still colder than today, enhances the interhemispheric and the land sea contrasts, favoring

the inland advection of moisture from the ocean onto the continent. Monsoon precipitations are enhanced and extend further north, whereas precipitation is depleted over the ocean (Fig. 5 c. and d.).

Compared to the PMIP3-CMIP5 IPSL-CM5A-LR simulation (Kageyama et al., 2013a), the PMIP4-CMIP6 IPSL-CM6-LR simulation produces a larger seasonal response in the northern hemisphere. It is mainly the result of colder winter conditions, which are also characterized by the fact that there is a DJF mean cooling

in the Arctic, instead of the warming due to reduced sea-ice extent in the PMIP3-CMIP5 simulation (Figure 4 a. and b.). Part of this offset between the two simulations results from the changes in trace gases in PMIP4-CMIP6 that contribute to a global annual mean cooling of all PMIP4 simulations (Brierley et al., 2020; Otto-Bliesner et al., 2017). The other part is attributed to regional climate features differences between the two model versions as discussed in Boucher et al. (2020). Several important differences appear

also in winter (DJF) in Eurasia, over the GIN (Greenland, Iceland, Norwegian) seas and over the Arctic (Fig 4 a and b., and Fig. 5 a. and b.).  The summer monsoon precipitation is more intense and in better agreement with data reconstructions from West Africa. It extends further north over Northern India and Pakistan. These differences translate in little differences in model skill when compared with paleoclimate reconstructions at the regional scale (Brierley et al., 2020), despite the large improvement in the simulations of the pre-

industrial or historical climates (Boucher et al., 2020).



### 3.2 Major differences between the dust sensitivity experiments

Changes induced by dust are of much smaller magnitude that the ones caused by insolation and trace gases (Fig. 6). The largest global radiative imbalance induced by the dust forcing and fast feedback at the top of the atmosphere is found in NODUST, and reaches about 0.1 W.m$^{-2}$ compared to MHREF (Fig. 1). In other words, completely removing dust induces a positive Dust ERP and a slight global warming, enhanced global evaporation and global precipitation, and larger sensible heat flux over land compared to the other simulations. The effect of dust on global model adjustments is thus very small, and smaller than the decadal fluctuations found for most variables (Fig. 1). This justifies that we estimate the 150 year mean annual cycle for the exact same period in all simulation with respect to their identical initial state.

Suppressing dust causes a winter warming of up to 2°C in West Africa and over the Tibetan Plateau as more shortwave radiation reaches the surface (Fig. 6a). A slight cooling of about 1°C is also simulated in the North Atlantic and a warming at the margin of sea-ice in the southern hemisphere. These large-scale changes in temperature affect the large-scale hemispheric gradients and lead to increase/decrease in precipitation in the northern/southern flank of the intertropical convergence zone (ITCZ) over the Atlantic and East Pacific Oceans (Fig. 6d). During summer, surface warming is limited to the coastal regions in West Africa and on the Tibetan Plateau (Fig. 6g). The African Monsoon precipitation is not affected, whereas precipitation is increased in North India and reduced over the tip of India (Fig. 6i). The largest precipitation differences with the reference mid Holocene simulation are found over the ocean where tropical Atlantic precipitation is increased and in the Pacific with a southward shift of the ITCZ.

Because of the much lower dust loads over North Africa in Albani0k compared to IPSL pre-industrial dust fields (Fig. 3b), the features from the Albani0k–MHREF difference (central column in Fig. 6) share similarities with NODUST–MHREF over Africa, e.g. in the DJF surface warming over west Africa (Fig. 6b vs 6a) and in precipitation changes in the tropical Atlantic, Africa and the Indian Ocean (Fig. 6e vs 6d). Note that a slight warming in the North Atlantic is simulated in both seasons (Fig. 6 b and h.). This is associated with a sea-ice reduction in the Labrador Sea. The drying of the African monsoon covers a broader area than in the NODUST case, and the rain belt is shifted southward in the Atlantic Ocean (Fig. 6k). On the other hand, dust loads in Albani0k are larger than IPSL over the Arabian Sea / Indian ocean, and lower over the





Indian subcontinent (Fig. 3). The Indian monsoon is homogeneously increased over India, in both cases
(Fig. 6k vs 6j).

The dust load is reduced in MH Albani's 6k compared Albani's PI dust (Fig. 2) (Albani6k-Akbani0k or f4 – f3, right column in Fig. 6). However, since the reduction is achieved with a minus/plus pattern between Africa and India, the simulated climate changes are different from NODUST when dust are suppressed everywhere. As in NODUST a cooling and drying expand from the north Atlantic to the Eurasian continent
during DJF (Fig. 6c). The cooling persists with small magnitude during JJA (Fig. 6i). Compared to NODUST-MHREF, a drying is simulated both for the African and the Indian monsoon regions, as well as a southward shift of the rain belt both in the Atlantic and the Pacific oceans (Fig. 6).

In order to identify the regions where the different sensitivity experiment exhibit the largest differences, we computed the root mean square difference between the four simulations over the whole annual mean cycle,
and highlighted the month for which the root mean square is at a peak. We compare these results to those computed for centennial variability, considering several non-overlapping 100-year mean annual cycles in the MHREF (f1) simulation. The results, shown in Figure 7, indicate that the differences in the tropic over land for temperature (Fig. 7 a and c) and over land and ocean for precipitation (Fig. 7 e. and g.) are indeed fully induced by the dust forcing, whereas in mid and high latitude the temperature pattern shares lots of
similarities with interannual variability both in magnitude and peak month (Fig. 7 b. and d.). For precipitation, variability is also large in the tropics, but with smaller magnitude and different peak months in most places than for the impact of dust forcing (Fig. 7 f. and h.). Figure 7 confirms that the maximum impact of dust on temperature in the tropics is found in West Africa, South of Lake Chad, and over the Arabian Peninsula during boreal winter from November to February (i.e. either month 11, or 12, 1 or 2), and
north of Lake Chad in boreal summer in July-August (month 7 or 8). For precipitation, the impact of dust is maximum during the West African summer monsoon season, in contrast with variability that peaks in Autumn (Fig. 7 g. and h.). Note that over the ocean peak differences between the effect of dust and variability share lots of similarities, except that larger and more homogeneous regions of similar peak differences are found for the dust signal (Fig. 7. g.). Maximum differences are mostly found during summer,
excepts in the west tropical Atlantic where the Spring season certainly reflect a slight shift in the timing of the migration or in the location of the Atlantic rain belt.



### 3.3 Moist static energy and atmospheric heat transport

The seasonal differences shown in Figure 6 counteract or amplify the mid Holocene signal depending on regions. Mid Holocene changes in atmosphere column moist static energy (MSE, Fig. 8) are consistent with the changes in temperature and precipitation and provide a more integrated view on the large-scale redistribution of energy associated to the insolation forcing. The major reduction of mid Holocene MSE induced by insolation during JF occurs in the Atlantic sector with maximum value over South America and Africa, whereas the major increase occurs over the Indian ocean and the West Pacific. The associated changes in total heat transport have thus a land ocean symmetry and an inter-basin asymmetry with increased MSE transport from the Indian Ocean to the Atlantic and Pacific and across the Pacific Ocean. The combination of reduced/increased meridional heat transport in the Atlantic/Pacific, leads to a net reduction of atmospheric heat exported from the tropical regional to higher latitudes in both hemispheres. It reaches about 0.18 PW at 40°N and around 0.15 PW at 35°S. During summer, the strong interhemispheric and land-sea differences induced by insolation and increased Afro-Asian monsoon enhances the energy export from the North Africa to the other ocean basins. This is mainly due to the large heating and specific humidity increase in this region. Note also that the export from Amazonia is also increased. These changes induce a net southward difference in total meridional heat transport across the equator that reaches about 0.3 PW.

As expected from the changes in temperature and precipitation shown in Figure 6, the energetic changes induced by the different dust forcings are of lower magnitude (Figure 8). However, both NODUST-MHREF and Albani6k-Albani0k, dust reductions are characterized by an increase in energy export from West Africa and South Atlantic subtropics to the north Atlantic in JF (Figure 8b). This feature is strengthened over the SH subtropical Atlantic. The latter is consistent with a reduced influence of the Saint Helen anticyclone which reduces the African monsoon (Figure 4b and5). Complete dust suppression in the NODUST-MHREF case therefore leads to a reinforcement of the insolation changes over the Asian sector and the enhanced role of the Asian monsoon in the MSE export (Figure 8b). It leads to an increase of the equator-to-pole heat transport in each hemisphere that reaches about 0.1 PW at 35°S, enhancing the role of the insolation forcing in the southern hemisphere and reducing it in the northern hemisphere. The situation is different with





Albani6k because of the East/West dust load pattern over Asia, which counteracts the NODUST effect

(Figure 8d). Because of this, there is almost no impact on the southern hemisphere meridional atmospheric

heat transport. The effect is similar to NODUST in the northern hemisphere, because it is mainly driven by

the Atlantic sector. The changes induced by Albani0k dusts reflect well the tripolar minus/plus/minus, dust

load pattern between West Africa, the middle East and East Asia (Fig. 3). During JF the reduced dust and

the associated warming over Africa reinforce the energy export to higher latitude as in the other two

simulations. However, in this case, the atmosphere over the Indian ocean becomes an energy sink. During

JA, changes in source and sink of atmospheric MSE counteract the effect of increased African monsoon,

while enhancing that of east Asian monsoon (Figure 8c).

These analyses reveal that even though the dust changes over Africa are large, the net impact on MSE

sources and sinks over the West African region with maximum dust changes is small. They show the

energetic response is driven by the changes in Atlantic sector when dust is suppressed or reduced, and that

the role of East Africa in the global energetics is affected by the difference in dust pattern between Africa

and Asia.

## 4 Global estimates and regional patterns of the radiative dust forcing

### 4.1 Estimation of the dust radiative forcing with two different methods

We estimated in section 2.3 dust ERP in MHREF, Albani0k and Albani6ka in order to compare the dust

radiative effect with those estimated by Albani and Mahoward (2019). We now go one step further by

estimating the instantaneous dust forcing separately in the SW and LW and considering as reference

MHREF, so as to identify the impact of dust reduction on the mid-Holocene climate (Table 3). We first

diagnose the dust instantaneous radiative forcing (IRF) associated with each sensitivity test following the

RFMIP protocol (Forster et al., 2016; Pincus et al., 2016). This is done by considering simulations with SST

prescribed to pre-industrial conditions, using the atmosphere-land surface component of the coupled

IPSLCM6-LR model used for the MHREF simulation. We thus performed three 30 year-long simulations

including a double call to the radiative scheme in order to diagnose the IRF associated to different dust

fields as done by Lurton et al. (2020) for historical and future climate. All simulations have mid-Holocene

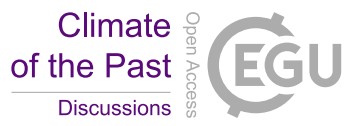

insolation and trace gases values. In a NODUST-type simulation with this model setup we diagnose the impact of suppressing dust compared to the IPSL reference pre-industrial (IPSL-PI) dust field used in MHREF. In an Albani0k-type simulation we diagnose the IRF of Albani0k dust field with respect to IPSL-PI reference dust field. Then in an Albani6k-type simulation the same procedure is applied to calculate the IRF of Albani6ka dust field with respect to the Albani0k dust field. The results (Table 3) are averages over

the last 10 years of each run. In all three simulations, the dust field used in the mainstream radiation call is reduced compared to the reference one (used in the $2^{nd}$ radiation call), so the annual global IRF-SW is positive: when dust is reduced/suppressed, the Earth system receives more radiation.

As expected, the annual global IRF-SW is maximum in the NODUST-type simulation, where the difference of dust conditions in $1^{st}$ vs $2^{nd}$ radiation calls is maximum: "no dust " vs. IPSL-PI dust field. IRF-SW

reaches 0.62 $W.m^{-2}$ in clear sky, and 0.41 $W.m^{-2}$ when clouds are also considered (Table 2). The effect of clouds partially offsets the direct effect of dust on albedo, which is mainly due to feedbacks from climate adjustment. The Albani0k IRF-SW, with respect to the same reference IPSL-PI, is much smaller : 0.09 $W.m^{-2}$ in clear sky vs 0.04 $W.m^{-2}$ in all sky. Since the Albani0k total dust load (14.6 Tg) is only ~20% smaller than the IPSL-PI dust load (18.7 Tg), and these cases essentially share the same size distributions and optical

properties, the differences mainly arise from the different horizontal patterns and altitude of dust in the atmosphere. Reduced mid-Holocene Albani6k dust load (12.8 Tg) compared to Albani0k results in an IRF-SW of the same order of magnitude as that of Albani0k vs. IPSL-PI, in both clear- and all-sky conditions. The IRF-LW is of smaller magnitude, and is negative, which also partially offset IRF-SW. As for IRF-SW the interaction with clouds slightly damps the clear sky effect. Interestingly the IRF long wave is very small

when comparing Albani6ka with Albani0k, compared to the two other case. Also, the ratio with IRF-SW is larger for Alabni0k. It stresses that the 3D-distribution in that atmospheric column and spatial distribution are 2 key factors for this effect. In particular, the difference in dust altitude between albani0k and Albani 6ka is small compared to the other cases.

In order to estimate the dust forcing and dust feedbacks in the adjusted coupled experiments, we also apply

the simplified partial derivative method developed by Taylor et al. (2007), valid for shortwave radiation (see method's description in the appendix). In this method, the absorption and scattering properties of an equivalent atmosphere are computed from the radiative fluxes and the surface albedo. It allows estimating





changes in these atmospheric and surface properties on the planetary albedo, and thereby on the net shortwave radiation at the top of the atmosphere. The comparison of the estimates using clear sky instead of all sky radiation variables allow to provide an estimate of the direct effect of dust, whereas the difference between the clear sky and the all sky estimate provides an estimate of clouds (cloud feedback) in the atmospheric budget on shortwave radiation, that reflect both the dust semi-direct effect and climate feedback.

The numbers obtained for clear sky with this method are quite close to the IRF-SW estimates (Table 3), indicating that, dust radiative perturbation is dominant under clear-sky conditions, and that interactions with clouds damp the total dust forcing (all sky). The estimates for all sky are however different (except for the NODUST case) suggesting that cloud distribution is different in the coupled simulations with SST adjustments. It however indicates that there is little impact of ocean feedbacks when moving from estimates in atmosphere alone simulations with prescribed modern SSTs to fully coupled mid Holocene simulations for clear sky. It also provides an indirect evaluation of the Taylor et al (2007) simplified method, that proves to be rather accurate for the case of dust radiative effects.

### 4.2    Regional differences in dust radiative forcing and feedbacks

As expected from the different dust load maps (Fig. 3), the small global numbers of Table 2 mask larger regional values and the differences in radiative pattern changes induced by the different dust fields (Fig 8 a and b). The dust loads in the atmosphere have a seasonal distribution resulting from regional meteorology (winds), soil moisture and vegetation characteristics (Marticorena, 2014; Shao et al., 2011; Tegen et al., 2002; Zender et al., 2003).  We only consider the SW clear sky forcing for illustration in Figure 9, since this term is dominating dust forcing. (Table 3). The comparison between IRF and the Taylor et al. (2007) clear sky estimates confirm that both global and regional estimates are similar (cf panels a and c in Fig 9 and 89b). The increase in surface radiation in regions where dust is suppressed or reduced (Fig. 9) is directly the fingerprint of the reduced effect of the shadow effect of dust on incoming solar radiation at the surface. It follows quite well the seasonal differences in dust loads. As anticipated, there is a pattern of plus and minus



values when considering Albani's 0K or 6k dust fields, characterized by increased dust loads along latitudes

from East Africa to the Middle East (Fig. 2).

At the top of the atmosphere over the Sahel-Sahara region, the radiative effect of dust is a patchy SW pattern between the Sahara and the Sahel regions, whereas the differences in surface radiation more directly represent the minus/plus pattern of the differences in dust load (Fig. 9). Further exploration using the Taylor et al. (2007) methods indicate that the compensation at TOA over the Sahara comes from the compensation

between the absorption and scattering effect of dust in the atmospheric column (Fig. 10a and 10 b). It leads to almost no changes in SW radiation at TOA over the Sahel-Sahara region during JF and reduced radiation in JJA. This is consistent with previous results obtained with the aerosol model INCA (Balkanski et al., 2007) and CESM1 (Albani et al., 2014), and in agreement with satellite-based estimates (Patadia et al., 2009) averaged over Sahara for present day. These studies highlight that the response over the Sahara is due

to dust SW direct radiative effect, and that it depends on the albedo of the underlying surface (Albani and Mahowald, 2019; Patadia et al., 2009). Note that the impact is not a direct effect of the surface albedo (not shown) but the effect of dust absorption and thereby of the relative balance between the changes in dust absorption and scattering that is affected by the surface condition (brightness).

In the regions directly affected by the change in dust the shortwave radiations in Figure 10 are very similar

to values found for clear sky conditions and reflect the direct impact of dust on the atmospheric radiative forcing (Fig. 9). Outside these domains, the values estimated with the Taylor et al. (2007) method reflect the feedbacks by atmospheric clouds induced by the changes in the atmospheric circulation and in sea-surface temperature and sea-ice cover (Fig. 10). In all three simulations changes induced by the seasonal evolution of vegetation LAI are small (not shown), but the albedo effect diagnosed on TOA radiation indicates it has a

small impact on the simulated climate in southern Europe and Northern tropics in JF, and over a small band in Sahel, reflecting less active vegetation where monsoon is reduced in this simulation. Figure 9 and 10 confirm that the largest TOA forcing and feedbacks are found in the Atlantic sector in JF for all cases, and that in JA they extend from Africa to Asia (Tibetan plateau) in NODUST compared to MHREF. The increase in net radiation in the Atlantic sector is compensated by a decrease in East Africa in Albani0k. The

change in SW radiation are of smaller magnitude in Albani6k when compared to Albani0k, and reflect a





mixture of the NODUST case and Albani0k case forcing and feedback estimates with MHREF as a reference.

The Taylor et al. (2007) method cannot be applied for longwave radiation. For longwave we provide an estimate of the global impact of dust forcing and associated feedbacks by estimating on the one hand the surface emission using the Planck function and surface temperature, and on the other hand the total atmospheric heat gain from the difference between outgoing radiation and surface emission. As expected regions with strong shortwave radiative effect induced by dust that are associate with a surface warming emit more radiation out of the atmosphere. This effect is partially compensated by lapse rate, humidity and cloud feedbacks, as well as dust effect on LW, in the atmospheric column (Fig. 10). These analyses stress that the amount of dust alone, is not sufficient to explain mid-Holocene dust forcing, but that the geographical distribution of the forcing is also very large and varies substantially from a dust model to another, with substantial differences between Africa and India.

## 5    Analyses of the impact of the different dust forcing on the Afro-Asian monsoon and Atlantic overturning

### 5.1    The African and Indian monsoon responses to insolation forcing

Here we discuss in more depth the latitude/altitude profiles over W Africa and India, to show how dusts affect atmospheric convection and precipitation, and how the dust effect acts on the insolation forcing (Fig. 11 and 11).

When comparing the MH and PI PMIP4 reference simulations, we see that, in response to the JA insolation forcing the temperature gradient is increased in the atmosphere at all levels over West Africa north of 20°N and over India over the Tibetan plateau (Fig. 11). Between 10°N and 20°N in Africa and 25°N and 35°N in India, temperature changes have a barocline structure, with colder temperature below 700 hPa and higher temperature above, which coincide with the regions with maximum change in monsoon precipitation. It is located to the south of the maximum increase in low level atmospheric moisture content in Africa and above it in India. In India the monsoon system is constraint by the Himalayan orography, and these changes are located just at the foothill of it. In both regions this pattern reflects the northward shift (or extent) of the monsoon rainbelt, and is associate to a local reinforcement of the southern branch of the Hadley cell that



contribute to export heat from the monsoon system to the south (Fig. 11 a) and Fig. 8 b). Further south the temperature is reduced over the whole atmosphere. Note that the increase atmospheric moisture content extends further north and is not limited to the rain belt regions. The moisture content increases also acts to

185 set up the large-scale change in the atmospheric circulation (Figure 8 b). The fact that moisture increases total atmosphere MSE far north on the continent is an important aspect in the northward shift of the rainbelt during mid Holocene. The changes in the atmospheric circulation are also characterized by a depletion of the atmospheric moisture content where subsidence is increased in the descending branch over the Atlantic Ocean, south of 5°N. Another difference between the African and Indian regions reflecting the regional

orography context is that the maximum vertical velocity, and precipitation is also located to the south of the minimum temperature in Africa, whereas it is located over the minimum temperature in India. Figure 12 (top) further highlights the mean changes between 10° and 20°N. This box has been chosen so as to show the barocline atmospheric structure in temperature and zonal wind changes with increase inland monsoon low from the Atlantic at low lever and the increase and upward shift of the 700 hPa African easterly jet

(AEJ) and increase of the 200 hPa easterly jet at upper levels. The changes in the AEJ are associate with the changes in meridional temperature gradient in the middle atmosphere as discussed in Texier et al. (2000). Figure 12 (top) also highlight the complex connection with the Indian monsoon in East Africa and over the Arabian see, with increased alimentation of the AEJ from above and an increased subsidence over the Arabian Sea where the subsiding flow joins the Indian monsoon south west surface flow around 60°E.

**5.2    Role of dust on the mid Holocene representation of the African and Indian monsoons**

The suppression of dust has an impact from the surface to about 500 hPa in both regions, with the maximum impact north of 10°N in African and north of 22°N in India. The suppression of dust coincides with a slight reduction of atmospheric moisture content over Africa (Fig. 11, NODUST-MHREF), and thereby a reduction of the large scale meridional MSE potential (Fig. 8d). Over India moisture is depleted over the tip

of India, but increased over the foothill of the Himalaya where upward motions and precipitation increase (Fig. 6 and 7). Changes in the atmospheric column for temperature partially counteract the effect of insolation with warmer temperature to the south and colder temperature to the north. Difference in atmospheric profiles between the insolation and dust forcing emerge in the regions where precipitation





change most (Fig. 11). The solar forcing acts at the surface whereas the suppression of dust acts both at the

surface by allowing more radiation to reach the surface and in the atmospheric column by changing the local absorption and scattering properties. Different mechanisms are thus implied to increase/decrease the monsoon strength. Insolation induce surface feedbacks that impact first the large-scale gradients, large scale dynamics and thereby moisture advection, which in turns is reinforced by local recycling (D'Agostino et al., 2019; Marzin and Braconnot, 2009; Zheng and Braconnot, 2013). Dust affect the surface temperature, but

also the temperature profile, relative humidity and atmospheric convection in the atmosphere, which in turn affects the large-scale circulation and the monsoon response. The results for Africa when dust is suppressed are consistent with previous findings on the role of dust. It was shown that the presence of dust reinforces the Saharan heat low in JA, with enhanced monsoonal flow and atmospheric humidity, and that the spatial structure of precipitation anomalies induced by dust shows an increase in monsoonal precipitations over the

Sahel only, which can be explained by the barocline structure of the atmosphere, that facilitates convection there rather than further inland (i.e. Miller et al., 2014). Over India this effect is counteracted by the increase warming of the southern flank of the Himalaya that acts as local elevated heat source in the Atmosphere and triggers upward flow and increase moisture convergence (Fig. 11). Therefore, the MSE energy source is increased over the Asian continent and reduce over Africa, enhancing the asymmetry between the two

monsoon responses. Differences between the insolation and the NODUST-MHREF case appear also clearly when looking at the zonal atmospheric circulation. Over west Africa, the reduces low level monsoon circulation is associate with a global warming, except between 800 and 600 hPa, where a cooling is related to a downward shift of the EAJ. The AEJ is however reduced in the above layers. There is also increased inflow in East Africa from the subtropical jet. The connection between the African and Indian systems over

the Arabian see sector is effect in the middle atmosphere but not at the surface. The latter correspond to the subsidence induce by increased Precipitation over India and the Bay of Bengal (Fig. 6 and 7).

The zonal mean patterns in temperature, humidity and wind changes induce by the Alabni0k dust on the MH climate or when comparing the effect of Albani6ka with Albani0k share similarities, with smaller magnitude compared to the Nodust case over west Africa below 600 hPa (Fig. 11). However, the surface warming

extends further north and there is a clear barotropic structure with warmer temperature between the surface and 600 hPa and colder one above. The situation is different for India, because of the zonal mean plus,



minus, plus pattern in atmospheric dust load (Fig. 2 and 10). In Albani0k, the reduced dust over the Himalaya foothill compared to MHREF favors the local increase of monsoon precipitation there, whereas the relative increase in dust over the Indian ocean and the middle East (Fig. 2c) acts to decrease the MSE

potential and the monsoon inland monsoon flow over East Africa, with a larger magnitude than in the Nodust case (Fig. 8c). When Albani6k dust is considered instead of Albani0k, mid Holocene dust are increased over India and the Tibetan plateau, which reduces the monsoon strength and the role of the heating source of the Southern flank of Himalaya (Fig. 11 Albani6k – Albani0k). The 10°N-20°N section indicates also differences in the atmospheric circulation over East Africa that are related to the E/W differences in

dust load. In particular the increase in dust in East Africa and Arabian Peninsula in Albani0k compared to MHREF has the same order of magnitude that the dust reduction in the west. It strongly reduces the monsoon flow in this region, and thereby humidity and MSE potential (Fig. 8). The westward extension around 500 hPa also show how the dust anomaly is connected between East and West Africa through the dust transport in the AEJ (Fig. 12), and contribute to increase dust in the middle atmosphere in west Africa

(Fig. 11). A similar pattern but with smaller magnitude is found when Albani6k dusts are considered instead of the Albani0k (Fig. 12). The reduced MSE and the changes in the atmospheric profiles induce an anomalous recirculating cell between the colder upper and the lower atmosphere with maximum subsidence around 30°E. This recirculation is connected with the walker circulation over the Atlantic. The comparison of Albani6k – Albani0k with the extreme NODUST-MHREF case shows that the dust pattern effect plays a

large role, and that depending on it, both the African and Indian monsoon are damped or there is an asymmetry in their responses.

### 5.3 Linkages with the Atlantic meridional overturning circulation

It is interesting to note that when dusts are suppressed (NODUST-MHREF) or reduced (Albani6ka – Albani 0k), the Atlantic overturning circulation is reduced (Fig. 1 and Fig. 13). We stressed in section 3.2 that the

pattern shares lots of similarities with interannual variability, and that it is marginally significant (Fig. 7, and Fig. 13). However, it is systematic, which requires further investigation on the connections with the dust forcing. The north Atlantic cooling in JA over the north Atlantic in NODUST-MHREF and Albani6k-Albani0k comes from the JF conditions (Fig. 6), and results from changes in atmospheric radiative





feedbacks induced by clouds (Fig. 10a), and from changes in the atmospheric and oceanic circulations, since

the direct dust forcing is not effective in these regions (Fig. 9a). It is amplified by sea-ice (not shown). The differences between in the three different cases imply similar dust forcing over West Africa and the eastern border of the tropical Atlantic, but the dust forcing in Albani0k-MHREF is negative in the West Atlantic (Fig. 9a). Because of this the net impact of Albani0k-MHREF dust forcing on the Atlantic trade winds is different than the one we get when suppressing (NODUST-MHERF) or reducing (Albani0k-MHREF)

atmospheric dust amount. This is also true in JA for the low-level anomaly winds (Fig. 12). The Albani6k-Albani0k dust reduction has a slightly different pattern than NODUST-MHREF complete dust suppression over East Africa and Asia. However, the magnitude is not large enough to counteract the large scale MSE differences resulting from a dust reduction (Fig. 9b and 9d). The NODUST-MHREF and the Albani6k – Albani0k MSE potential maps and transport share similar pattern in the Atlantic sector, corresponding to an

increase atmospheric interhemispheric transport over the Atlantic sector in JF and increase subtropical to north Atlantic atmospheric heat transport in JA (Fig. 9). There is thus an increase in the atmospheric Atlantic heat transport to the north that is compensated by a decrease in Atlantic oceanic heat transport. The Atlantic changes in MSE potential are associated with a JF northward shit and JA southward shift of the Atlantic ITCZ, that contribute to export energy from the regions with maximum MSE changes. This increased the

convergence of the NW trade wind around 5°N in JF and reduces its strength in the west subtropical north Atlantic. During JA, it reduces the trans equatorial southwest trade wind. These changes in winds connect the atmospheric response to the ocean wind forcing and the reduction of the ocean northward surface mass transport (Fig.12).

## 6    Discussion and conclusion

We analyzed the impact of dust reduction on the mid Holocene climate considering both an extreme case where dust is suppressed, and a mid-Holocene dust reduction from interactive dust-climate simulations following the PMIP4 dust protocol (Otto-Bliesner et al., 2017). We use as reference the new mid-Holocene simulations with the IPSLCM6-LR version of the IPSL model (Boucher et al., 2020), in which the spatial distribution of PI dust is prescribed from the IPSLCM6-LR CMIP6 PI simulations (Lurton et al., 2020). The





comparison of this simulation, after the adjustment phase, with the previous simulations using the

IPSLCM6A-LR version of the model (Dufresne et al., 2013) (Kageyama et al., 2013a; Kageyama et al.,

2013b) highlights a few key regional differences arising from new developments in all the climate

components and the higher resolution. For instance, Figure 14 summarizes that in North America, Europe

and West Africa the differences between the two model versions (grey and red bars in Fig. 14) are larger

than the mean differences between 100-year periods arising from internal variability (hashed red bar in Fig.

14).  However, despite the large improvement of model climatology of PI and historical climate (Boucher et

al., 2020), these differences do not concur to a better agreement when compared with Bartlein et al (2011)

paleoclimate reconstruction from pollen and macro fossil data (Fig.13). This results from the size of the

regions that do not necessarily allow to detect small improvements in part of the region. It also results from

systematic model biases, such as a wrong balance of the annual mean precipitation changes (MAP) between

Western North America and Europe, or an underestimation of precipitation changes in the Sahel region, and

this even when the large uncertainties on paleo reconstructions are considered. The major improvement is

for the colder coldest month temperature (MTCO), whereas the results for mean annual temperature (MAT)

seem less satisfactory, in particular with regards to its sign (Fig. 14). This figure also highlights that over

these large regions the changes in temperature or precipitation induced by dust are marginally significant

and the results of different dust forcing are very similar to the reference version. This somehow contrasts

with the robust differences with the previous model (Fig. 14).

We found systematic differences induced by dust between the sensitivity mid-Holocene dust experiments.

We test the impact of different dust distributions for the pre-industrial climate comparing a simulation with

an imposed dust preindustrial distribution from the CESM1 model with the reference IPSL simulations for

which dust is prescribed from a previously run dust simulation produced by the interactive INCA aerosol-

chemistry model. The comparison with the standard mid Holocene simulation reveals that the largest

difference with the reference mid-Holocene simulation are induced by dust load pattern, which is

characterized by an atmospheric dust reduction in Africa and an increase over the Middle East. These

differences in dust pattern also have an impact on the climate response to mid-Holocene dust reduction. It

thus stresses that the simulated dust pattern in the PI control simulation is also an important factor to



consider when analyzing the effect of dust in a different climate. It also implies that the effect of the mid-Holocene dust reduction cannot be simply deduced from the extreme no dust case.

These conclusions come from detailed analyses of the dust radiative forcing and effect of dust on the Indian
and African monsoons. The forcing differences induce both changes in local and global circulation resulting from the redistribution of MSE between source and sink regions. The fact that the Atlantic sector is most affected in all simulations compared to the other oceans therefore reflects the dominant effect of dust over Africa and the fact that the dust pattern has a direct impact on induced atmospheric thermodynamics and dynamics changes. However, the magnitude and exact pattern over the tropical Atlantic depend on the
teleconnection induced by the differences in dust load over the Middle East and Indian sector. It also stresses that differences in these forcing patterns, as well as the differences in dust representation and interaction with atmospheric radiation need to be accounted for to be able to properly interpret the sensitivity of different climate models to dust forcing. The response to our first two questions is thus that the dust induces a forcing that is about 10 times smaller than the insolation forcing, but that it can be as large as
the insolation forcing in very specific regions such as Africa where changes are the largest. The balance between the induced changes in West African and Indian monsoons depends on the dust pattern, which might explain inconsistent responses between models in their simulated relative changes between the monsoon rain intensity and location over India and Africa. A key difference between the two regions is also associated with orography and the fact that the warming/cooling over the southern flank of the Himalaya can
induce different regional response over India than the one expected from radiative changes over the Tibetan plateau, whereas for Africa differences are led by dust properties and the linkages between atmospheric dust radiative effect and surface albedo.

Our results confirm previous results over Africa for the mid-Holocene period (Albani and Mahowald, 2019; Hopcroft and Valdes, 2019; Liu et al., 2018; Pausata et al., 2016; Thompson et al., 2019). At least part of the
differences in the reported results are attributable to different dust optical properties and whether absorption by dust of longwave radiation was included. When the longwave effects are included, the presence of large particles in describing dust particle size distributions then becomes important. This is in line with studies using present day climate conditions, that show how models using more absorbing optical properties tend to be associated with positive DRE at the Top of the Atmosphere (TOA) and increased Sahel precipitation,







whereas models using less absorbing optical properties tend to produce the opposite effects (e.g. Albani et al., 2014; Balkanski et al., 2007; Lau et al., 2009; Miller et al., 2014; Strong et al., 2015). Recent observational constraints (e. g. Di Biagio et al., 2019; Kaufman et al., 2001; Sinyuk et al., 2003) indicate less absorbing dust optical properties than older estimates (e. g. the OPAC dataset from Hess et al., 1998), and the IPSL model dust radiative properties are consistent with satellite-based estimates (Patadia et al.,

2009; Song et al., 2018).

Several studies also suggest that the increased monsoon induced by dust reduction during mid-Holocene is effective only when the mid Holocene increase in vegetation over the Sahel-Sahara region is considered (Pausata et al., 2016). This is not the case here, so that other conclusions on the effect over Africa could be obtained with interactive vegetation, with an increase instead of a decrease in monsoon extent and

precipitation over Africa.

Another important point is the linkages with the Atlantic Ocean. Part of those depends on the large-scale dust pattern, in particular from dust transport and feedbacks with the location of the ITCZ and changes in precipitation in the West Atlantic. Our results show that depending on this region being affected or not in terms of winds and precipitation, there is a feedback on the Atlantic thermohaline circulation, which is

slightly reduced in the simulations presented here. This needs further investigation, because the signal is small and projects on the north Atlantic centennial variability. Very long simulations or ensembles would be needed to fully assess this effect.

Since the Atlantic sector is the one that is most affected by the impact of the dust forcing whatever the pattern, paleoclimate reconstructions over this sector seem particularly relevant to evaluate the results. This

is why we selected the well sampled regions in Eastern North America and Western Europe in Figure 14. However, a full assessment of the model realism can be only by done considering both the realisms of dust representation and forcing pattern in the reference climate (here pre-industrial) and the mid Holocene changes. Therefore, future explorations need to consider the consistency of dust and land surface properties, and put more emphasis on dust outside Africa to properly assess pattern changes and the balance between

the different atmospheric teleconnections and the way by which they induce changes in MSE large scale gradients.





## 7    A. Appendix: simplified method to estimate dust forcing and feedback for short wave radiation

The Taylor et al. (2007) method is only valid for the estimation of climate forcing for shortwave radiation. A first step consists in diagnosing how the surface albedo, atmospheric absorption and atmospheric scattering affect the planetary albedo for each climatic period.

For each simulations the atmospheric absorption μ is estimated as:

$$\mu = \alpha_{p+}\left(SWsi/_{SWi}\right)(1-\alpha_p) \tag{1},$$

and the atmospheric scattering γ as:

$$\gamma = \frac{\mu - \left(SWsi/_{SWi}\right)}{\mu - \alpha_s\left(SWsi/_{SWi}\right)} \tag{2},$$

where $\alpha_p$ and $\alpha_s$ represent respectively the planetary and the surface albedos, and $SW_i$ and $SW_{si}$ the incoming solar radiation at the top of the atmosphere (insolation) and at the surface. The planetary and surface albedos are computed from the downward and upward SW radiations.

The partial derivative to obtain the effect of the change in absorption between simulation 1 and simulation 2 is thus estimated as :

$$\alpha_p(\mu_2, \gamma, \alpha_s) - \alpha_p(\mu_1, \gamma, \alpha_s) \tag{3}$$

The estimate of the changes in atmospheric scattering and in surface albedo are estimated following the same methodology.

### Data availability

The PMIP3 and PMIP4 coupled simulations presented here can be found on the CMIP6 ESGF database. Model outputs follow the required ESGF standards and CMIP6 data request (https://pcmdi.llnl.gov/CMIP6/).

### Author contribution.

PB, SA, YB and MK designed the study and experiments. SA, YB and AC prepared the dust files. PB run the simulations, preformed most analyses and wrote the manuscript with the contributions of all authors. AS performed the IRF analyses using the RFMIP protocol and the corresponding simulations. SA, OM, JYP and



MK also contributed respectively to the analyses of forcing, thermohaline circulation, surface temperature

and precipitation maps, and model-data comparisons.

The authors declare that they have no conflict of interest

**Acknowledgements.**

This work is a contribution to the PMIP4-CMIP6 project (https://pmip.lsce.ipsl.fr/) and is supported by the

JPI-Belmont PACMEDY project (N ° ANR-15-JCLI-0003-01). It was undertaken in the framework of the

Institut Pierre Simon Laplace Climate Modeling Centre. As such it benefited from the French state aid

managed by the ANR under the "Investissements d'avenir" program with the reference ANR-11-IDEX-

0004-17-EURE-0006. The CMIP6 project at IPSL used the HPC resources of TGCC under the allocations

2016-A0030107732, 2017-R0040110492, and 2018-R0040110492 (project gencmip6) provided by GENCI

(Grand Équipement National de Calcul Intensif). It also benefited from the ESPRI (Ensemble de Services

Pour la Recherche l'IPSL) computing and data centre (https://mesocentre.ipsl.fr) which is supported by

CNRS, Sorbonne Université, Ecole Polytechnique and CNES and through national and international grants.

S. A. acknowledges funding from the European Union's Horizon 2020 research and innovation program

under the Marie Skłodowska-Curie grant agreement 708119, for the project "DUSt, Climate, and the Carbon

Cycle" (DUSC3).

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





| Simulation name | Initial state | Dust file | length |
|---|---|---|---|
| MH adjust | Year 1850 of IPSLCM6A-LR PI simulation | PI-IPSL | 350 years |
| r1i1p1f1 (f1) | Year 2190 of MH adjust | MHREF | 550 years |
| r1i1p1f2 (f2) | Year 2190 of MH adjust | NODUST | 300 years |
| r1i1p1f3 (f3) | Year 2190 of MH adjust | Albani0k | 300 years |
| r1i1p1f4 (f4) | Year 2190 of MH adjust | Albani6k | 290 years |

Table 1. Characteristics of the different experiments, in terms of initial state, 3D dust distribution and length of the simulation. Note that (f4) is 290 instead of 300-year long. The last 10 years should have been run on a new computer, with the risk to obtain non-continuous results due to the new running environment on the new computer. Since the simulation was long enough for our purpose, we decided to stop it at year 290.




| Experiment | Dust load (Tg) | Dust Effective Radiative Perturbation (W m$^{-2}$) | Dust Absolute Effective Radiative Perturbation (W m$^{-2}$) |
|---|---|---|---|
| f1 (MHREF) | 18.7 | -0.10 | \|0.80\| |
| f3 (Albani0k) | 14.6 | -0.11 | \|0.69\| |
| f4 (Albani6k) | 12.8 | -0.05 | \|0.56\| |
| CESM1 PI (Albani and Mahowald, 2019) | 20 | -0.-06 | \|0.27\| |
| CESM1 MH (Albani and Mahowald, 2019) | 18 | -0.03 | \|0.23\| |

**Table 2**. Global budgets of simulated dust-related variables. Dust Effective Radiative Perturbation (W m$^{-2}$)
is the globally average net (SW+LW) TOA effective radiative perturbation estimated from the first 50 years
of the simulation and calculated as a difference compared to NODUST. It includes thus initial model
adjustment as discussed in Albani et al. (2019). Dust Absolute Effective Radiative Perturbation (W m$^{-2}$) is
similar, but considering the perturbation irrespective of the sign. The cases CESM1 PI and MH refer to the
original simulations (Albani and Mahowald, 2019) that were taken as the PMIP4 dust reference fields (Otto-
Bliesner et al., 2017). Note that this is the only table for which the comparison is done to NODUST, since it
highlights the impact of dust in the climate system, and not the effect of dust on the mid-Holocene climate
as it done in the core of the manuscript.





| Method | RFMIP-IRF (instantaneous forcing) at TOA from double calls to radiation scheme in forced simulations (W.m$^{-2}$) | | | Taylor et al. (2007) forcing approximations at TOA in coupled simulations (W.m$^{-2}$) | | |
|---|---|---|---|---|---|---|
| Dust field(s) | NODUST vs. IPSL-PI | Albani0k vs. IPSL-PI | Albani6k vs. Albani0k | NODUST Vs IPSL-PI | Albani0k vs IPSL-PI | Albani6k Vs Albani0k |
| SW Net Clear sky | 0.62 | 0.09 | 0.07 | 0.64 | 0.09 | 0.06 |
| SW Net All sky | 0.41 | 0.04 | 0.05 | 0.33 | 0.09 | -0.01 |
| LW Out Clear sky | -0.09 | -0.03 | -0.004 | - | - | - |
| LW Out All sky | -0.07 | -0.024 | -0.001 | - | - | - |

Table 3: SW net and LW outgoing dust radiative forcing (W.m$^{-2}$) at TOA in clear sky and all sky conditions as estimated from the different simulations. The RFMIP-IRF method diagnoses the forcing online at each model radiation timestep, with respect to a reference (using a double call to the radiation scheme), here in simulations without oceanic adjustment. The Taylor et al. (2007) method diagnoses the total radiative

forcing of dust (forcing and feedback) from the fully coupled experiment after ocean adjustment. This method does not allow to diagnose LW IRF. See section 4.1 for details on the radiative forcing estimates.



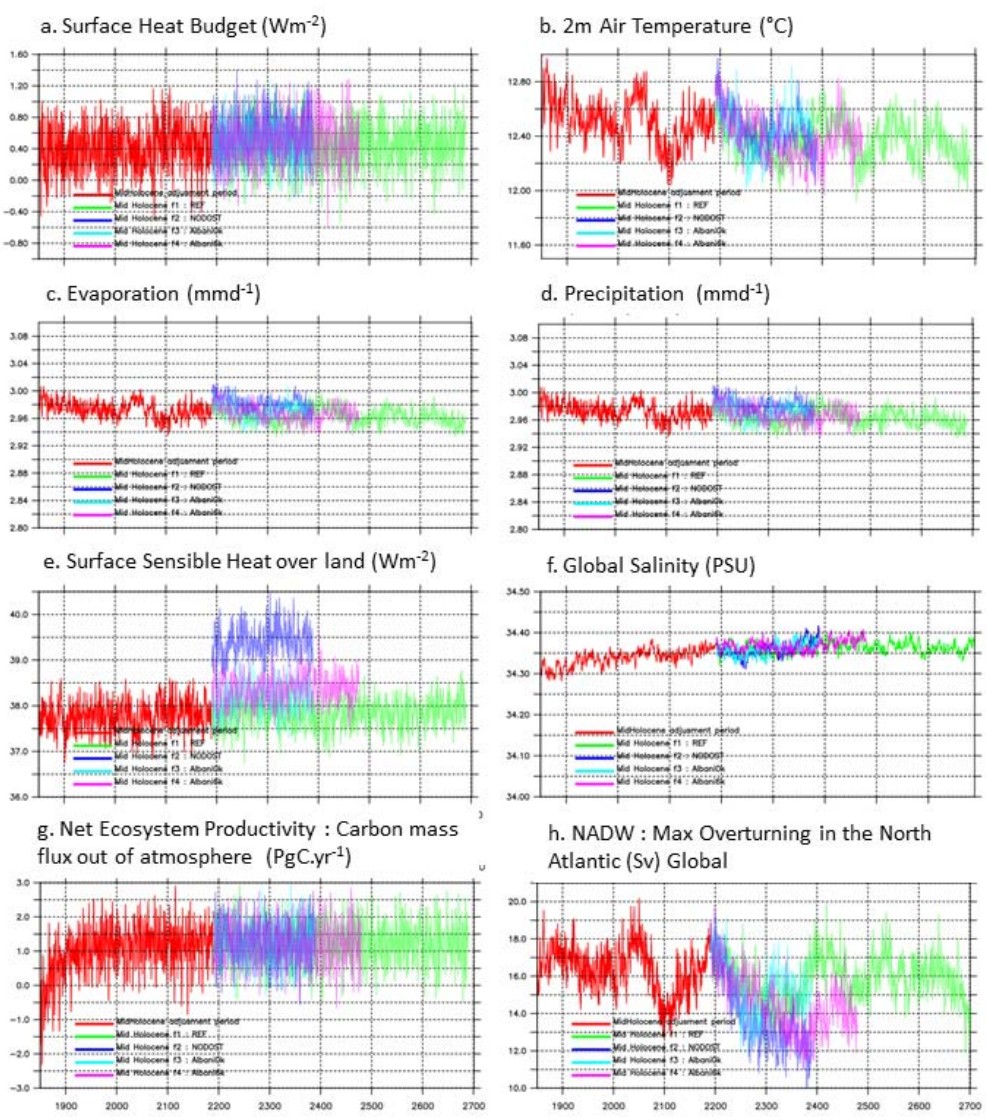

Figure 1. Temporal evolution of a subset of atmosphere, ocean and land surface variables for the different simulations: a. Surface heat budget (Wm$^{-2}$), b. 2m air temperature (°C), c. Evaporation (mmd$^{-1}$); d. Precipitation (mmd$^{-1}$); e. Surface sensible heat over land (Wm-2); f. Global salinity (PSU); g. Net Ecosystem Productivity (PfCyr-1) and h. Maximum Overtunring in the North Atlantic (Sv). In each plot, the red curve represents the adjustment phase from the Preindustrial condition when the midHolocene forcing (Earth's orbit and trace gases) is switched on. The green curve correponds to the reference mid Holocene simulation, whereas the other curves stand for the different sensitivity



tests to the dust forcing, starting from the same initial state than the referenc midHolocene
simulation.

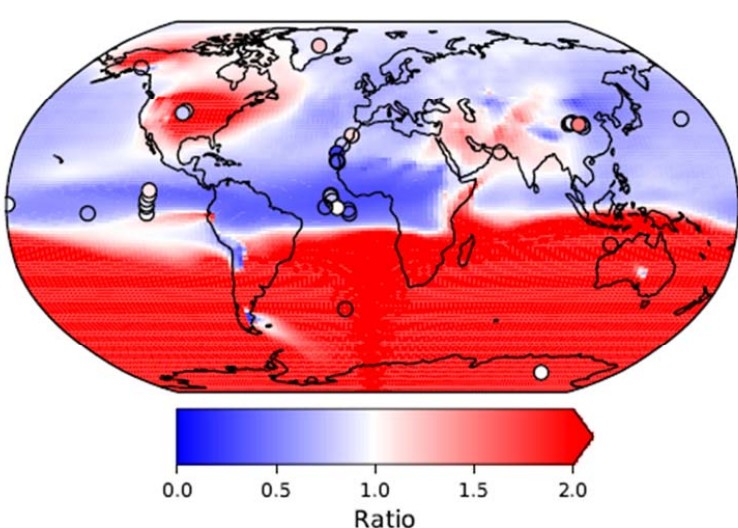

Fig. 2 : Ratio between Albani et al. (2015) mid-Holocene and pre-industrial dust loads (color map) and ratio of dust deposition as estimated from dust reconstruction (REF) (colored circles)





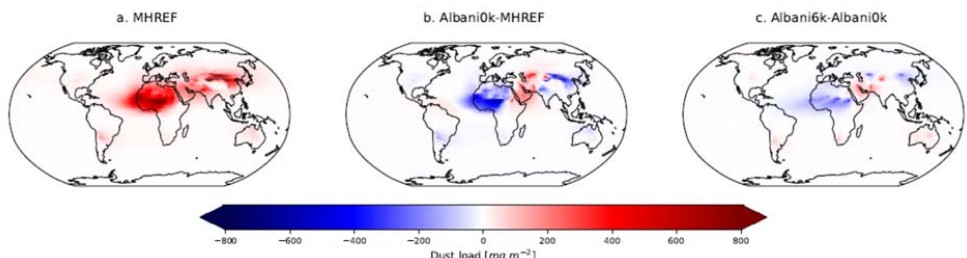

Figure 3. Comparison of the annual mean dust loads (mg.m$^{-2}$) from the different simulations. a. reference version with the Present day IPSL dust load as used in the reference PD-IPSL mid-Holocene simulation. b. differences between the Albani's and IPSL PI dust load and c. differences between Albani's MH and PI dust load.

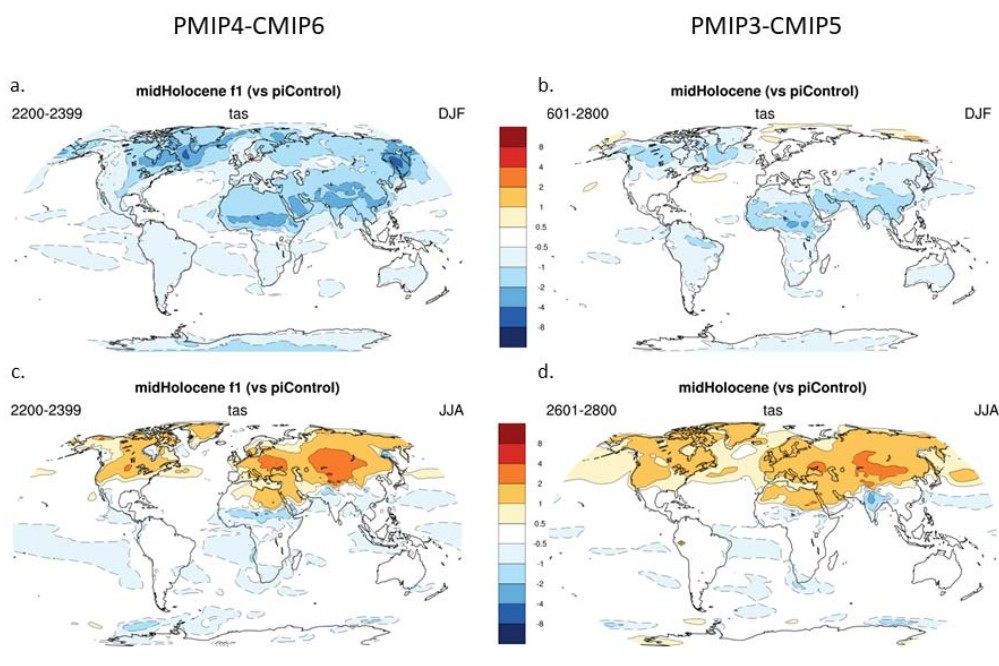

Figure 4: 2m air temperature differences (tas, °C) between the mid Holocene and the preindustrial
climates as simulated for December-January-February (DJF) (a,b) and June-July-August (JJA) (c,d) in
the IPSL-CM61-LR PMIP4-CMIP6 simulations (a,c), and the IPSL-CM5A-LR PMIP3-CMIP5 simulations
(b, d).



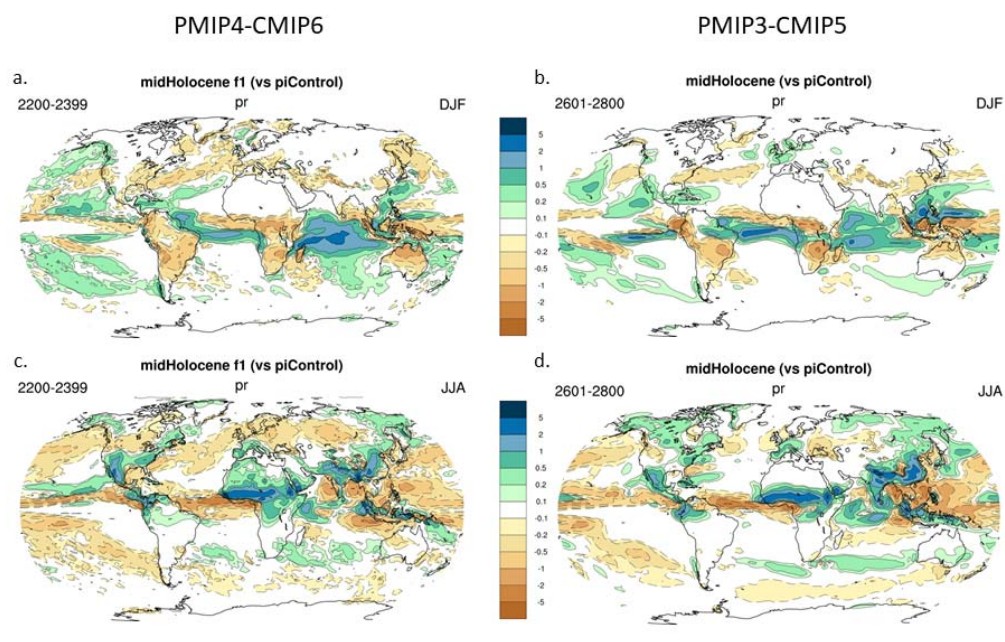

Figure 5 : Same as Fig. 3 but for precipitation (pr, mm.d⁻¹)

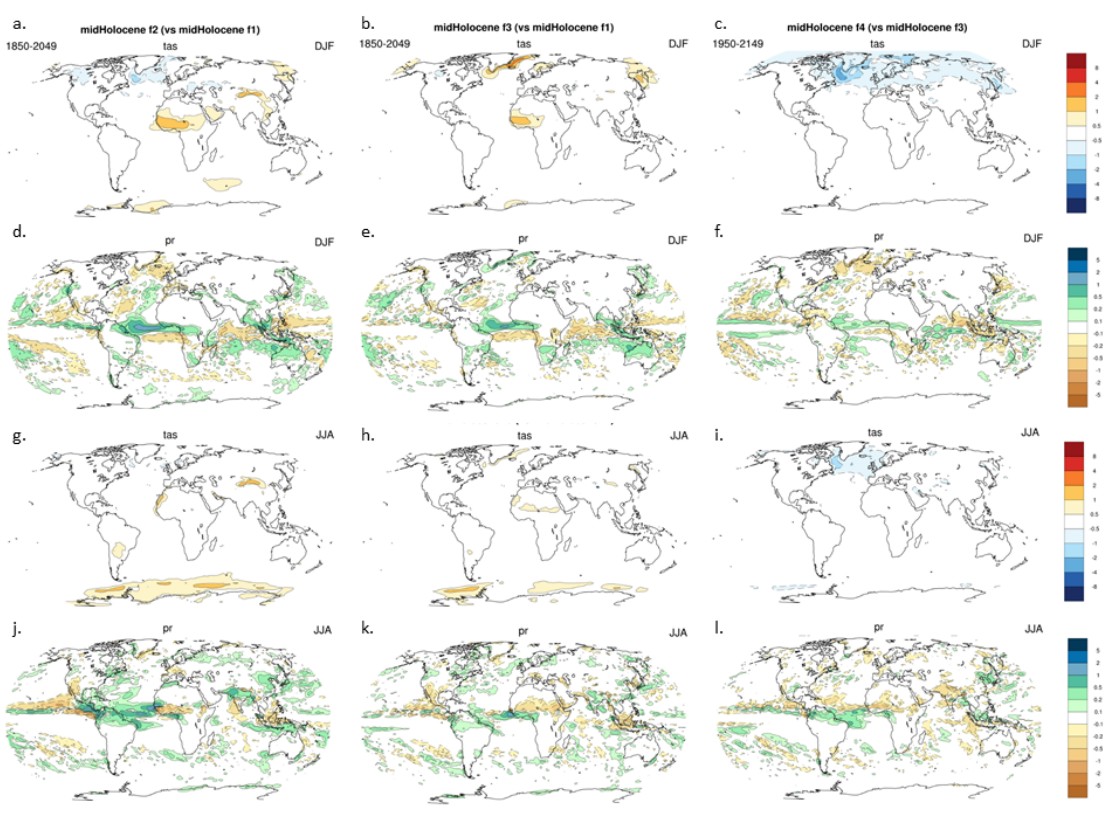

Figure 6. Impact of the different dust forcings on the simulated mid-Holocene climate, considering in the first two rows (a., b., c., d., e., and f.) December-January-February (DJF) differences and the last two rows (g., h., i., j., k., and l.) mean June-July-August (JJA) differences in temperature (tas, °C, a., b., c., g., h. and i) and in precipitation (pr, mm. d$^{-1}$ , d., e., f., j., k. and l. ) , so as to highlight in the first column (a., d., g. and j.) the effect of suppressing dust (f2 vs f1, left), in the middle column (b., e., h. and k.) the effect of replacing IPSL preindustrial dust by Albani's preindustrial dust (f3 minus f1) andin the the last column (c., f., i. and l.) the effect of mid-Holocene dust considering Albani's dust files (f4 versus f3).





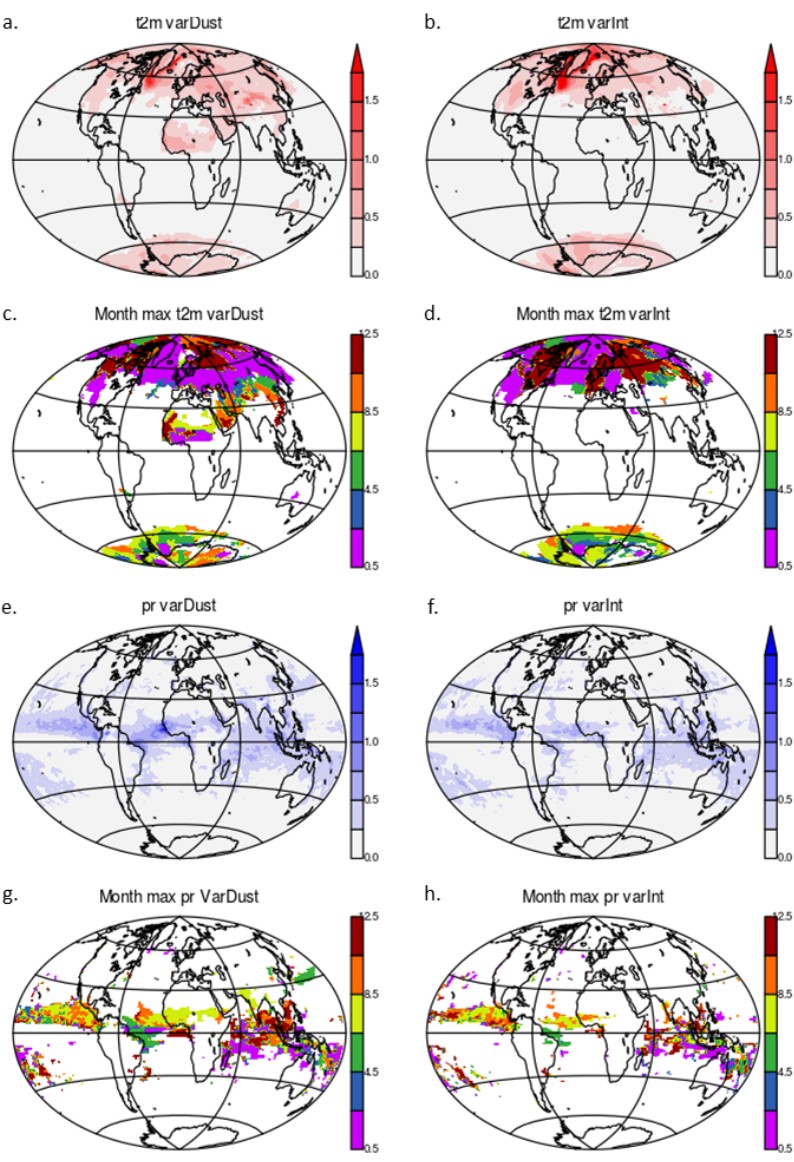

Figure 7. Identification of the regions where differences are maximum between a., c., e and g. the mid Holocene dust sensitivity experiments compared to b., d., f., and h. regions where centennial variability is maximum considering 2m air temperature (a., b., c. and d.) and precipitation (e., f., g. and h). The different analyses correspond to a. and e. mean root mean square differences between the four mid-Holocene simulations with different dust forcing, b. and f. mean root mean square differences between non-overlapping 100-year periods of the reference mid-Holocene simulation f1, and a., b., g. and f. months for which the root mean squares for the different analyses is maximum. 2m air temperature is expressed in °C and precipitation in mm.d$^{-1}$.

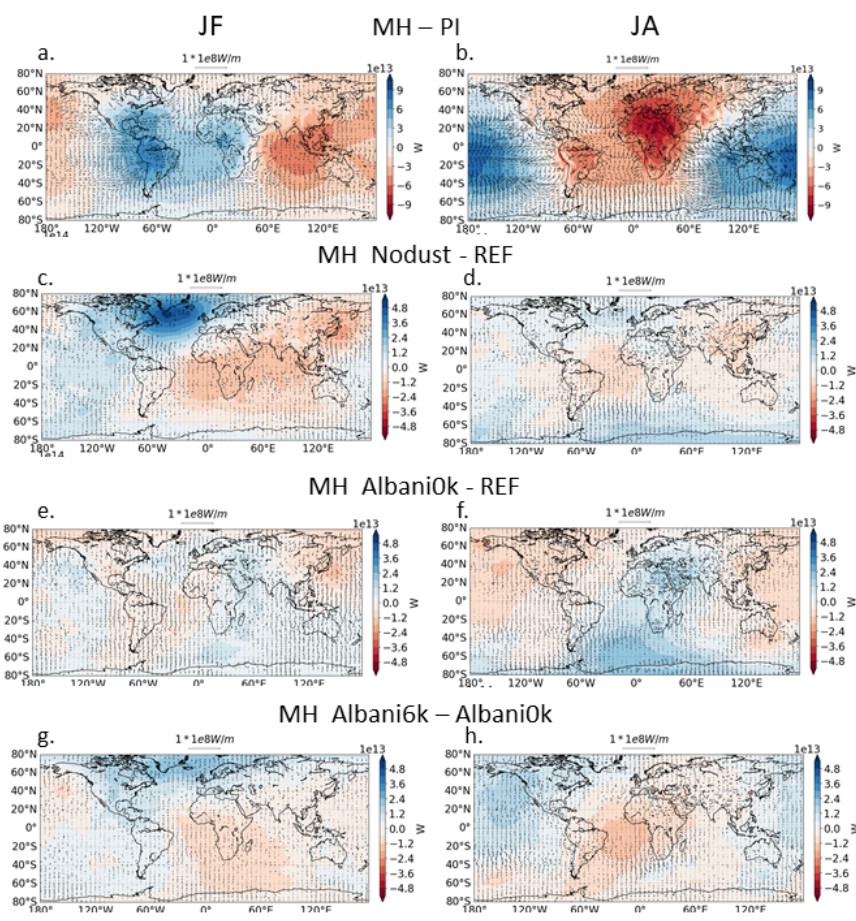

Figure 8 : Difference of moist static energy integrated over the atmospheric column (W) and associated moist static energy transport (W.m$^{-1}$) for a. January-February (JF)and b. July-August (JA) averages of the difference between the mid-Holocene reference simulation (MH REF) and the preindustrial control simulation (PI) , c. JF and d. JA averages of the Mid-Holocene Nodust (MH Nodust) and reference simulations, e. JF and f. JA averages of the Mid-Holocene Alabani0k dust Albani0k) and reference simulations, and g. JF and h. JA averages of the Mid-Holocene Albani6k and the Mid-Holocene Albani0k simulations.





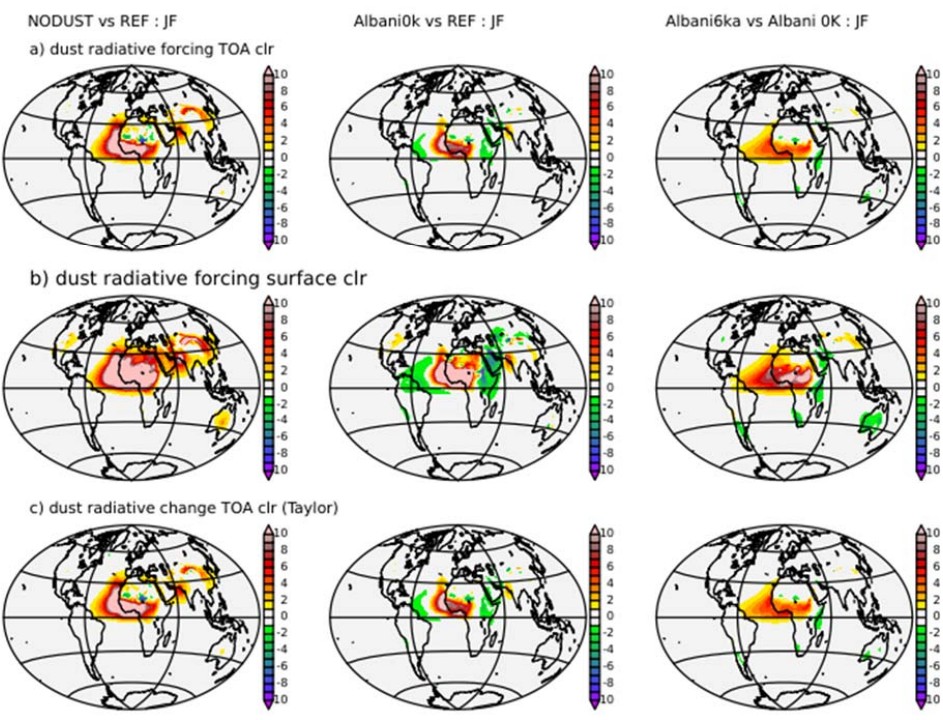

Figure 9a :January-February mean radiative forcing (W.m$^{-2}$) estimated from online atmosphere alone simulations following the RFMIP protocol at a) the top of the Atmosphere (TOA), b) at the surface and c) using the simplified Taylor et al. 2009 approach for clear sky values at TOA in the coupled simulations. The left and middle columns represent respectively the difference between the NODUST (left), and Alanani0k (middle) estimates compared to the Mid-Holocene reference simulations, and the right column the difference between the Albani6k (right) and the Albani0k estimates. All fields are positive down to directly reflect the radiative budge at TOA or at the surface. See text for details on the different estimates.



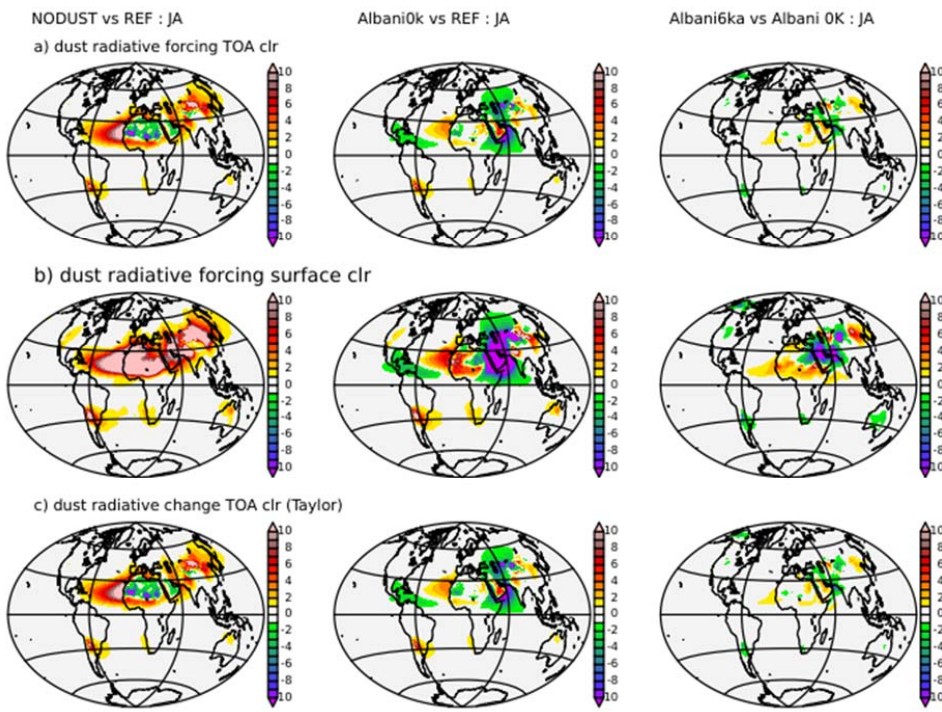

Figure 9b: Same as figure 8a, but for July-August averages





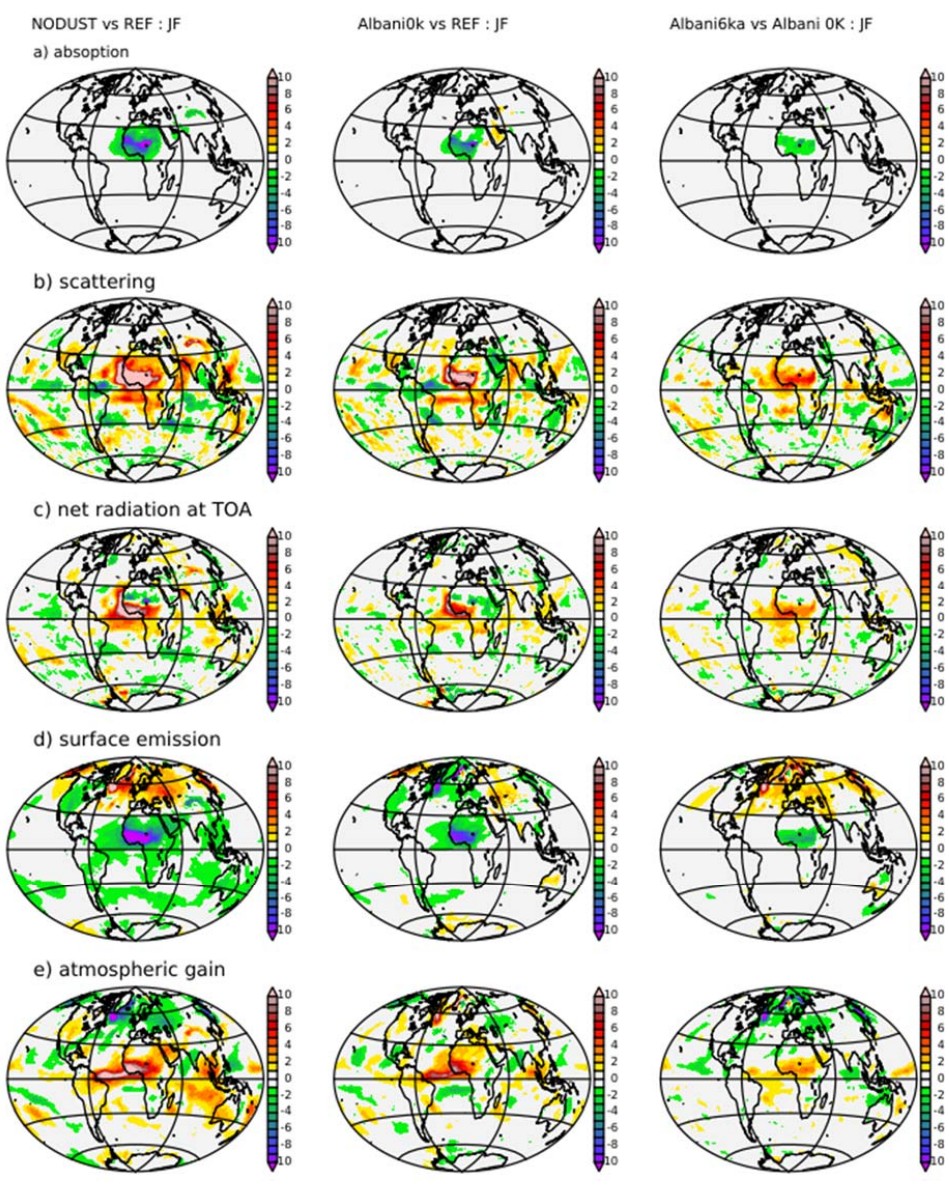

Figure 10a: Impact of dust on top of the atmosphere radiation considering the decomposition of Taylor et al (2009) for shortwave radiation and a) the effect of changes in atmospheric absorption, b) the effect of atmospheric scattering, c) the change in net radiation (SW +LW) at the top of the atmosphere (TOA), and for long wave the estimate of d) the change in surface emission due to temperature (Planck emission) and e) the effect of changes in the atmospheric heat gain, which represent the aggregated effect of changes in atmospheric lapse rate, change in atmospheric water vapor and changes in clouds. From left to right the different panels show the results for the NODUST,



the Albani0k and the Albani6k cases, with the first two compared to the mid-Holocene reference simulation and the last one to the Albani0k mid-Holocene simulation. All terms are positive downward to directly reflect the relative impact of each of these terms on the atmospheric radiative budget et the top of the atmosphere. The change in net radiation at TOA is also the sum of all the other terms, given that the effect of changes in surface albedo, due to changes in leaf area index, is negligible (not shown).



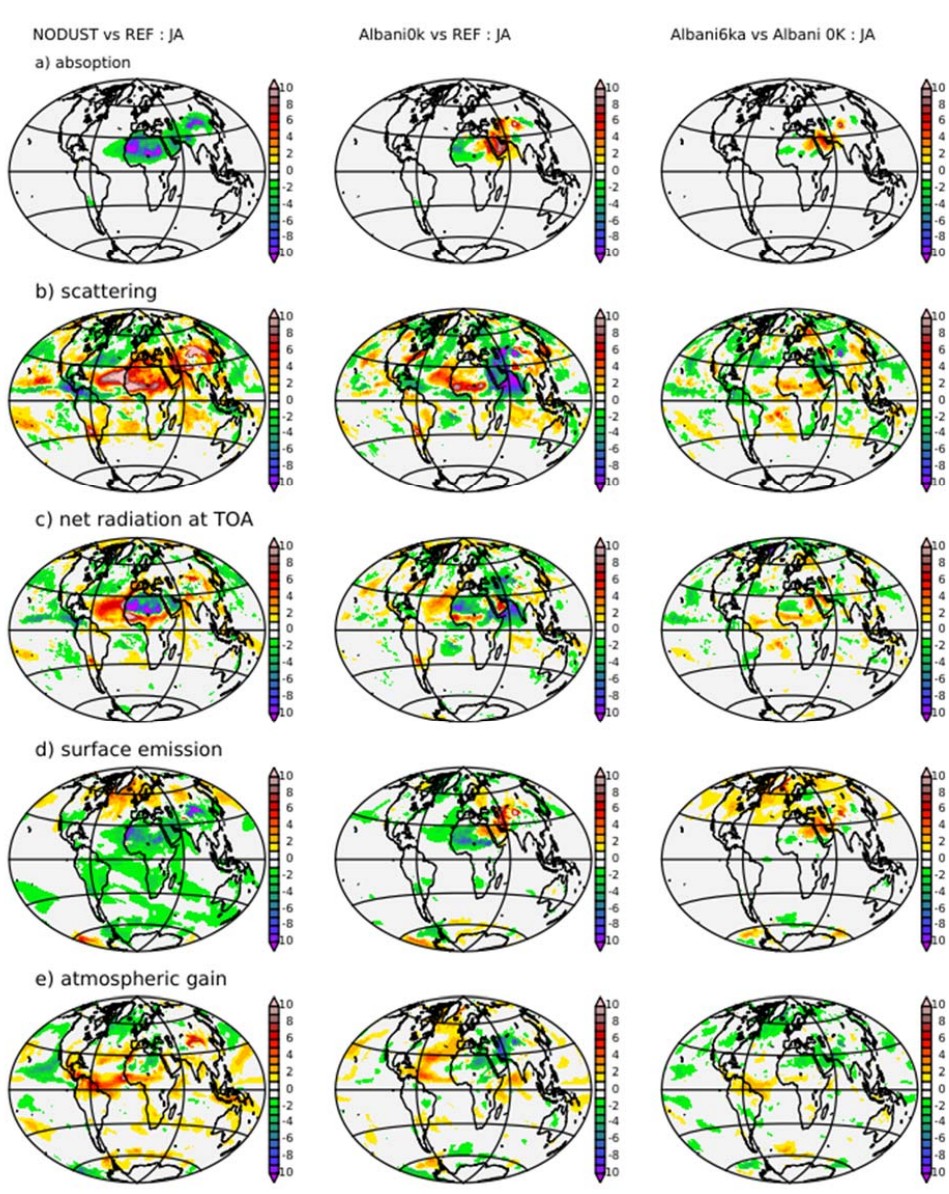

Figure 10b: Same as figure 9a, but for July-August averages

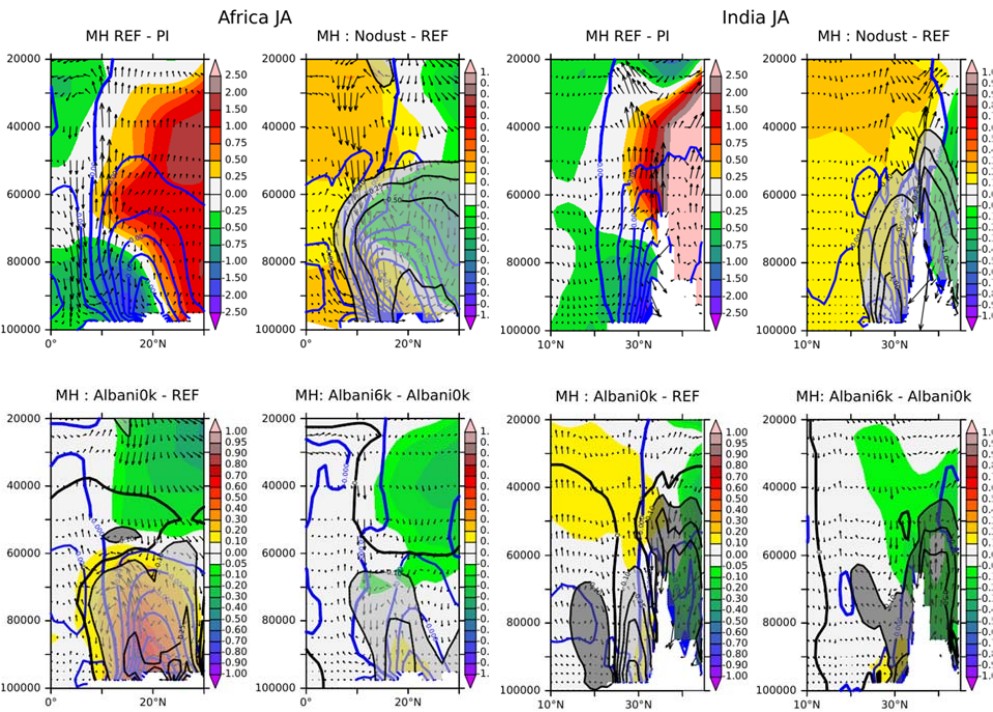

Figure 11 : Zonal averages of the July-August mean atmospheric circulation over the a) West African (10°W to 10°E) and b) the Indian (70°E to 90°E) monsoon regions showing the effect of the mid-Holocene insolation (top left), and the effect of NODUST (top righ), Albani0k dust (bottom left) and Albani6k dust (bottom right) on the atmospheric temperature (°C, color scale), the meridional winds (with a -1 factor imposed on the vertical component in hPa/s, to have more intuitive vision of upward and downward motions), the specific humidity (kg/kg, blue contours), and the dust concentrations (black contours, plotted on top of light grey where the dust content is reduced and of dark grey where dust content is increased). For each monsoon regions the top right panel has different scales, which reflect the larger impact of the insolation forcing, and no dust contour, which accounts for the fact that there is no change in dust between the mid-Holocene reference simulation and the preindustrial simulation.

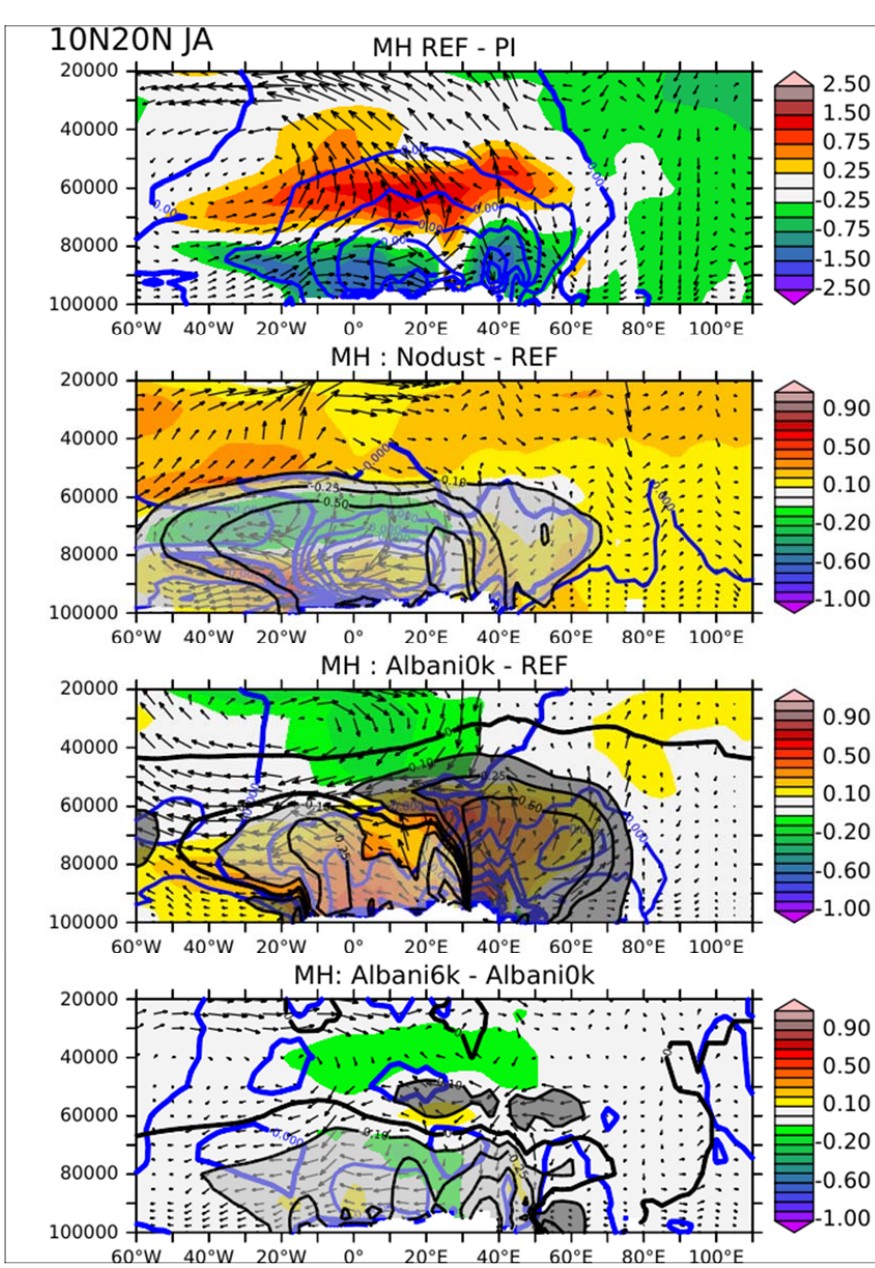

Figure 12: As in Figure 10, but the meridional circulation averaged between 10°N to 20°N. These changes reflect the changes at the northern edge of the boreal summer monsoon rain belt in Africa.



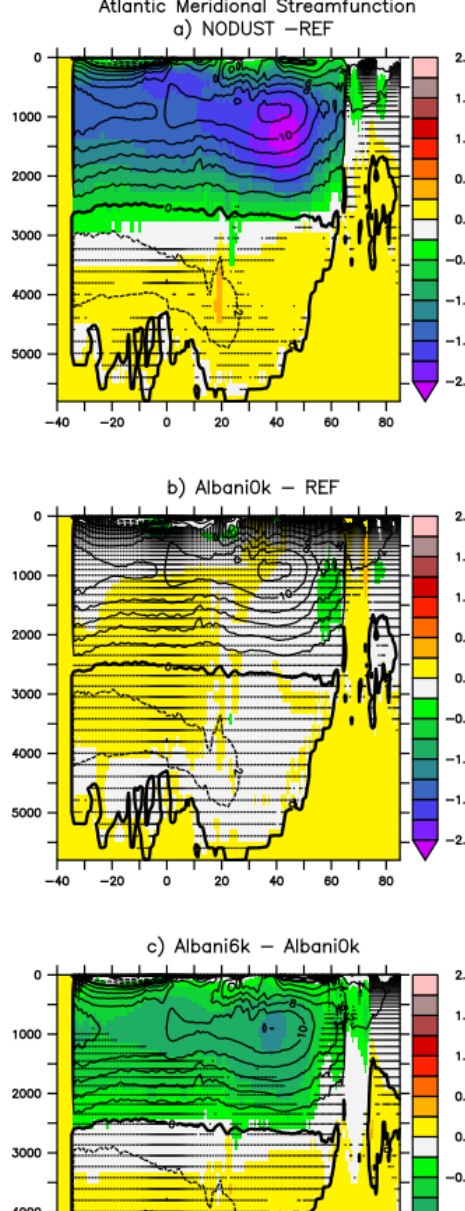

Figure 13 : Differences in the meridional Atlantic overturning circulation(Sv) as a function of latitude
(°) and depth (m)  showing the effect of a) NODUST, b) Albani0k dust, and c) Albani6k dust on the



mid-Holocene simulations. For a) and b) the reference is the reference mid-Holocene simulation, whereas for c) the reference is the Albani0K mid-Holocene simulation. Stippling indicates values that are smaller than the centennial variability of the midHolocene reference simulation, where the differences are considered as non-significant.

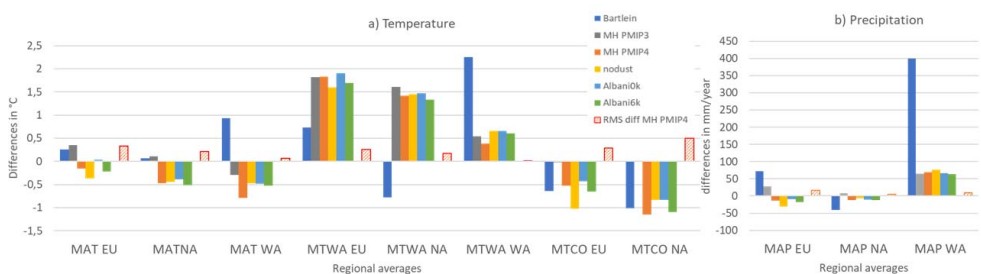

Figure 14. Comparison of the simulated changes with Bartlein et al. (2016) temperature and precipitation reconstructions over Europe (EU), North America (NA), West Africa (WA). The different indicators are the Mean Annual Temperature (MAT, °C), the Temperature of the warmest month (MTWA), the temperature of the coldest month (MTCO, °C), and the Mean Annual Precipitation (MAP, mm.yr$^{-1}$)