# Peer review of "Impact of dust in PMIP-CMIP6 mid-Holocene simulations with the IPSL model"

_Climate of the Past, 2020_

## Referee Comment (RC1) · Anonymous Referee #1 · 17 Dec 2020

The manuscript presents the results from several simulations with IPSL model regarding the effects of dust on mid-Holocene climate. The authors have used a suppressed dust by completely removing the dust forcing, and a reduced dust by considering what happened during mid-Holocene humid period when there were less dust emissions. The authors have provided the detailed analyses of the dust radiative forcing on mid-Holocene climate, with particular focus on western African monsoon and Indian monsoon region. They conclude that taking into reduced dust in mid-Holocene account has minor improvements for the simulated climate when comparing with the reconstructions. However, they emphasise that the dust pattern matters, which determine the changes in atmospheric thermodynamics and dynamics. The comprehensive test on dust forcing in this work is important to clarify the issues on neglecting the reduced

dust in simulating mid-Holocene climate by using the same dust forcing as that in PI. The manuscript is well written and has provided clear message to the modelling community. For the final publication please consider the below minor comments.

1. For the dust forcing the authors only focus on the direct and semi-direct radiative forcing, but did not mention the indirect effects. The indirect effect is important for the monsoon region where the deep connection occurs. If the indirect effect can not be estimated in these experiments, some discussion would help the reader to understand why it has not been considered, and by neglecting the indirect effect, how much monsoon response could be underestimated.

2. In Fig2, Albani0k and Albani6k dust show increased dust in middle East, it would be helpful if the authors provide the information what cause this increase instead of refer to a reference. Reference for reconstruction in fig2 caption is missing.

3. On the model-data comparison presented in Fig14, the authors have mentioned that the dust forcing is not significant on regional climate change, given that the model results show large disagreement with the reconstructions (even opposite in sign), some explanation for the possible reasons would be helpful.

4. The figure quality needs to be improved, for example, it is difficult to observe the moist static energy transport vectors in Fig8. And in Fig11 and Fig12, it is difficult to see the numbers labeled in contours.

5. There are some typos throughout the manuscript, need to be carefully checked and corrected, some examples below (Line number is not shown complete, only show 2 numbers):

P6, L52, MMD and SD, provide what do they stand for.

P9, L21, "on ESG, should be ESGF

P10, L62, "It extends further north over Northern India and Pakistan", should be further west?

P11, L76, "and over the Tibetan Plateau", actually the region is north of the Tibetan Plateau, should be Gobi desert region, it does make sense with reduced dust in Fig2.

P11, L82, Tibetan Plateau should be Gobi desert.

P11, L83, Fig .6i should be Fig. 6j, in Fig6j should mention that large difference in precipitation also in Indian monsoon and East Asian monsoon region.

P12, L10, "...with interannual variability", should be centennial variability?

P13, Section 3.3, talked about the meridional heat transport in PW in different latitude, which figures are these numbers referring to?

P17, L53, Tibetan Plateau should be Gobi desert.

P20 L27, "associate with global warming", why global warming in this case?

P20 L27, "the reduces low level...", should be the weakening of low level...

P20 L28, "EAJ", should be AEJ

P22 L 66, "between in the ...", remove in

P23 L 03, "colder coldest", remove colder

Table 2, -0.-06, should be -0.06?

Fig7, how to read the peak month? It is better to mark 1, 2, 3...12 with different colour.

---

## Referee Comment (RC2) · Anonymous Referee #2 · 14 Jan 2021

Braconnot and the coauthors study the role of dust in the mid-Holocene climate using an IPSL earth system model. They use sensitivity experiments to discuss how dust affects the climate. The issues discussed are important for understanding the mechanism of climate change and fruitful for the paleoclimate modelling community. Although the experiments are well designed to clarify the issues discussed, I feel that the manuscript is not well organized and there are careless inconsistencies here and there. I would also like to know more about the analysis of the biogeochemical aspects of dust effects, since the earth system model is used. Although they discussed the relationship of dust with AMOC, I would also like to know if there could be a correlation of dust with the suppression of ENSO during mid-Holocene. Providing insights on the role of dust on future climate change would enhance the value of the paper. Also, many

of the figures are not of sufficient quality for publication at CP. Hence, I suggest major revisions before publication. Each comment is listed below.

L.13 mid-Holocene: the expression of mid-Holocene is not consistent in the manuscript (mid Holocene, midHolocene), standardize it.

L. 29: Introduction discusses only the issue of dust in mid-Holocene but a broader context would attract wider readers. Discuss the role of dust on climate of the past and future and why mid-Holocene is important among the different times.

L.71 mid mid-Holocene –> mid-Holocene

L. 73 Can you explain briefly what INCA-IPSL model is?

L. 99 Two expressions are mixed in each experiment in the manuscript, making it difficult to understand: MHREF, NODUST, Albani0k, Albani6k and f1, f2 . . .. Use one type. I prefer MHREF . . ..

L. 115 a slight reduction — I have an impression nearly 30 % of reduction is more than "a slight reduction"

L. 253 DJF: It is not clear that the calendar adjustment (Otto-Bliesner et al. 2017 GMD) is applied for the seasonal analyses throughout the study.

L. 259 Several important differences appear – Explain what the important differences how does they mean.

L 261: The summer monsoon precip. . . – Judged from Fig 5, the precipitation over Sahara seems greatly improved than previous modelling studies. Can you explain how much improved from the previous models and why? I could not distinguish pronounced improvement of the IPSL model at Fig. 8 of Brierley et al. CP 2020. Please discuss on it.

L 261: data reconstruction – give references

L 342: JF – why do you choose January-February instead of DJF? The same for JA.

L 597: Bartlein et al. 2011: Kaufman et al. 2020 Scientific Data, https://doi.org/10.1038/s41597-020-0445-3 provides narrower constraints than Bartlein.

L 598 paleoclimate reconstruction from pollen and macro fossil data (Fig.13). – Figure 13 does not explain paleoclimate reconstruction.

Fig. 14 is only discussed in the Discussion section. But I think the Discussion and Conclusion should be a summary of the entire paper. I suggest that Discussion and conclusion to start from Line 608. And add one section beforehand for from Line 585 to Line 607.

Line 615: mid-Holocene dust reduction –> mid-Holoicene

Table 1: If you call experiments by dust file, reform the table. For clarity, it is recommended to put MHREF, NODUST, etc. in the first column (simulation name), and the ripf code and f1 . . . can be in the last column. The same for Table 2.

Table 2: -0.-06 –> -0.06 ?

Figure 1: labels and legends are too small to read. Figure 1 (b): There exist clear negative trend of global mean temperature. This means that the experiments are not equilibrium. In such cases, I think it is important to align the time periods of the original experiment and the analysis of the branched group of experiments. In addition to the lack of equilibrium, there is another reason why you should take the same period. From (h), the 100-year variability of the AMOC is so large that if you take different periods, you may see differences in the AMOC state due to different phases.

In Figure 1 some experiments are only shown for 200 years while in table 1 all experiments are shown for more than 290 years.

Fig. 2: Add proper reference at REF.

Figure 3: Make the figures bigger and fill in the gaps to make them easier to read.

Figure 6: How about a narrower range of temperatures? The colours in the diagram could be clearer and easier to read.

Figure 7: c, d, g, h – the highest and the lowest colors are difficult to distinguish.

Figure 9: c panels are almost the same with a. There are quite a lot of figures in this study. The comparison of a and c could be at supporting figures?

Figure 10: provide unit.

Figure 11: This is very busy figure. In order to get the point across, you need to make it easier to read. At least, I cannot read numbers at blue lines. Some of the numbers on color bars are half hidden. Give a, b, c . . . for each panel.

Figure 12: As in Figure 10 – > As in Figure 11

Figure 13: Should yellow color start from 0.25 Sv? Mask non-oceanic grids to black or gray. Southern boundary could be 35S? Uniformly dotting the stippling entire panel improves visibility.

L. 690: According to the data policy of climate of the past, the authors are requested to upload the data and codes for the analyses to a reliable data repository (or as supplement materials) so that anyone can reproduce the data in figures and tables. The authors are also requested to provide DOIs for the data on the ESGF.

---

## Author Comment (AC1) · 15 Feb 2021

Responses to reviewer 1

*The manuscript presents the results from several simulations with IPSL model regarding the effects of dust on mid-Holocene climate. The authors have used a suppressed dust by completely removing the dust forcing, and a reduced dust by considering what happened during mid-Holocene humid period when there were less dust emissions. The authors have provided the detailed analyses of the dust radiative forcing on mid-Holocene climate, with particular focus on western African monsoon and Indian monsoon region. They conclude that taking into reduced dust in mid-Holocene account has minor improvements for the simulated climate when comparing with the reconstructions. However, they emphasise that the dust pattern matters, which determine the changes in atmospheric thermodynamics and dynamics. The comprehensive test on dust forcing in this work is important to clarify the issues on neglecting the reduced dust in simulating mid-Holocene climate by using the same dust forcing as that in PI. The manuscript is well written and has provided clear message to the modelling community. For the final publication please consider the below minor comments.*

*1. For the dust forcing the authors only focus on the direct and semi-direct radiative forcing, but did not mention the indirect effects. The indirect effect is important for the monsoon region where the deep connection occurs. If the indirect effect can not be estimated in these experiments, some discussion would help the reader to understand why it has not been considered, and by neglecting the indirect effect, how much monsoon response could be underestimated.*

In the current version of the IPSLCM6 model only the first aerosol indirect effect is parameterized, for soluble aerosol species; therefore dust, that is treated as insoluble, does not contribute to this effect in our simulations. As the reviewer suggests, this effect may be important, and future developments will take this into account. It is however difficult to quantify it.

*2. In Fig2, Albani0k and Albani6k dust show increased dust in middle East, it would be helpful if the authors provide the information what cause this increase instead of refer to a reference. Reference for reconstruction in fig2 caption is missing.*

Albani0k and Albani6k represent the interpolation into the IPSLCM6-LR model framework of the prescribed dust fields from the CESM simulations described in Otto-Bliesner et al. 2017 and Albani et al. 2015 and 2016. These prescribed dust fields were obtained using an "assimilation" process, so that the causes of Holocene variability cannot be inferred from the simulations themselves. For the middle East / central Asia region the observational constraints that were used are on two marine sediments records from the Arabian Sea (Pourmand et al. 2004, 2007), and the GISP2 ice core record from Greenland (Mayewski et al. 1997), based on information on dust provence (Bory et al., 2003).. There is indeed scarcity of relevant data and significant uncertainty for this region, although newer work may improve our understanding of paleodust variability in this region. These precisions are included in the revised manuscript.

Bory, A. J.-M., Biscaye, P., and Grousset, F. E.: Two dis- tinct seasonal Asian source regions for mineral dust deposited in Greenland (NorthGRIP), Geophys. Res. Lett., 30, 1167, doi:10.1029/2002GL016446, 2003.
Mayewski, P. A., Meeker, L. D., Twickler, M. S., Whitlow, S., Yang, Q., Lyons, W. B., and Prentice, M.: Major features and forcing of high-latitude northern hemisphere atmospheric circulation us- ing a 110,000-year-long glaciochemical series, J. Geophys. Res., 102, 26345, doi:10.1029/96JC03365, 1997.

*3. On the model-data comparison presented in Fig14, the authors have mentioned that the dust forcing is not significant on regional climate change, given that the model results show large disagreement with the reconstructions (even opposite in sign), some explanation for the possible reasons would be helpful.*

Unfortunately, this would require interesting and extensive analyses, which is out of scope for this paper. The IPSL model has improved a lot over this sector concerning model developments and representation of midlatitude climate and Atlantic Ocean (I.E Boucher et al. 2020) The expectation would then be that it also helps to get better model-data agreement for this region for the mid-Holocene, which doesn't seem to be the case.

Boucher, O., J. Servonnat, A. L. Albright, O. Aumont, Y. Balkanski, V. Bastrikov, S. Bekki, R. Bonnet, S. Bony, L. Bopp, P. Braconnot, P. Brockmann, P. Cadule, A. Caubel, F. Cheruy, F. Codron, A. Cozic, D. Cugnet, F. D'Andrea, P. Davini, C. de Lavergne, S. Denvil, J. Deshayes, M. Devilliers, A. Ducharne, J.-L. Dufresne, E. Dupont, C. Ethé, L. Fairhead, L. Falletti, S. Flavoni, M.-A. Foujols, S. Gardoll, G. Gastineau, J. Ghattas, J.-Y. Grandpeix, B. Guenet, L. Guez, E. Guilyardi, M. Guimberteau, D. Hauglustaine, F. Hourdin, A. Idelkadi, S. Joussaume, M. Kageyama, A. Khadre-Traoré, M. Khodri, G. Krinner, N. Lebas, G. Levavasseur, C. Lévy, L. Li, F. Lott, T. Lurton, S. Luyssaert, G. Madec, J.-B. Madeleine, F. Maignan, M. Marchand, O. Marti, L. Mellul, Y. Meurdesoif, J. Mignot, I. Musat, C. Ottlé, P. Peylin, Y. Planton, J. Polcher, C. Rio, N. Rochetin, C. Rousset, P. Sepulchre, A. Sima, D. Swingedouw, R. Thieblemont, A. Traoré, M. Vancoppenolle, J. Vial, J. Vialard, N. Viovy, and N. Vuichard, Presentation and evaluation of the IPSL-CM6A-LR climate model, Journal of Advances in Modeling Earth System, https://doi.org/10.1029/2019MS002010.

*4. The figure quality needs to be improved, for example, it is difficult to observe the moist static energy transport vectors in Fig8. And in Fig11 and Fig12, it is difficult to see the numbers labeled in contours.*

We will improve Figure 8 as it suffered from the conversion in pdf format. For both figure 11 and 12 we need to fix a pyferret bug, and have already spend time trying to overcome it. We fully agree that these Figures need improvement and will proceed to improve them.

*5. There are some typos throughout the manuscript, need to be carefully checked and corrected, some examples below (Line number is not shown complete, only show 2 numbers):*
Thank you for mentioning these typos. We corrected them.

*P6, L52, MMD and SD, provide what do they stand for.*
MMD is Mass Median Diameter. SD is Standard Deviation . We address these definitions in the text

*P9, L21, "on ESG, should be ESGF*
Done

*P10, L62, "It extends further north over Northern India and Pakistan", should be further west?*

The text now reads : "It extends further to the northwest of India and Pakistan"

*P11, L76, "and over the Tibetan Plateau", actually the region is north of the Tibetan Plateau, should be Gobi desert region, it does make sense with reduced dust in Fig2.*
This is right and we adjusted the text accordingly

*P11, L82, Tibetan Plateau should be Gobi desert*.
This has been corrected

*P11, L83, Fig .6i should be Fig. 6j, in Fig6j should mention that large difference in
precipitation also in Indian monsoon and East Asian monsoon region.*
This will be adjusted in the revision, considering also only JF and JA averages instead of DJF and JJA,
to have a consistent approach throughout the manuscript.

*P12, L10, "...with interannual variability", should be centennial variability?*
Corrected

*P13, Section 3.3, talked about the meridional heat transport in PW in different latitude,
which figures are these numbers referring to?*

It refers to figure 8. Figure numbers have been added in the text

*P17, L53, Tibetan Plateau should be Gobi desert.*
Corrected

*P20 L27, "associate with global warming", why global warming in this case?*
Thank you we suppress gobal

*P20 L27, "the reduces low level...", should be the weakening of low level...*
Corrected

*P20 L28, "EAJ", should be AEJ*
Corrected

*P22 L 66, "between in the ...", remove in*
Corrected

*P23 L 03, "colder coldest", remove colder*
Done

*Table 2, -0.-06, should be -0.06?*
Corrected
*Fig7, how to read the peak month? It is better to mark 1, 2, 3...12 with different colour.*

We started with a map where the 12 months were represented with different colors, but such
colored figure was too busy to convey the messages we were looking for. The point for this figure is
to highlight if the major differences due to dust forcing occur in winter, summer or other inter-
seasons, and if they are different or not in terms of annual timing compared to centennial variability.
This is the best we managed to do with the software we are using and available color maps, with the
request of having similar colors for month 11-12 and 1-2 that are part of the same season.

---

## Author Comment (AC2) · 15 Feb 2021

Responses to reviewer 2

*Braconnot and the coauthors study the role of dust in the mid-Holocene climate using an IPSL earth system model. They use sensitivity experiments to discuss how dust affects the climate. The issues discussed are important for understanding the mechanism of climate change and fruitful for the paleoclimate modelling community. Although the experiments are well designed to clarify the issues discussed, I feel that the manuscript is not well organized and there are careless inconsistencies here and there. I would also like to know more about the analysis of the biogeochemical aspects of dust effects, since the earth system model is used. Although they discussed the relationship of dust with AMOC, I would also like to know if there could be a correlation of dust with the suppression of ENSO during mid-Holocene. Providing insights on the role of dust on future climate change would enhance the value of the paper. Also, many of the figures are not of sufficient quality for publication at CP. Hence, I suggest major revisions before publication. Each comment is listed below.*

We would like to thank the reviewer for comments that help to improve the manuscript. We have tracked the inconsistencies and corrected the figures. We also adjusted the place of the model-data comparison. We respond point by point to the comments below starting with a response to the general comments and questions about additional analyses the reviewer would like to see in this study.

Response to the general comments.

The reviewer would like to know more about the biogeochemical cycle in these simulations. Since dust is not fully interactive in this version of the model the changes in biogeochemistry are driven by changes in ocean dynamics and physics. The marine biogeochemical cycle is out of the scope of this paper and would require engaging other specialists in the analyses and add time consuming analyses for which we do not have the resources at the moment.

We do not address the questions on ENSO because we focus on the annual mean cycle and mean energetics. We mention centennial variability because of large variability in the model in the north Atlantic and the fact that the differences between the dust experiments exhibit a signal in the same region. An important part of the analyses and the way to perform them was really to make sure that we do not attribute the difference in the north Atlantic to a dust effect when in practice it results from this multidecadal to centennial variability. This emerges from figure 1 showing the low frequency variability of the AMOC for example at a function of time at the centennial time scale. To address the reviewer's concern we made a rapid computation of the magnitude of the SST(°C), seasonal cycle and SST DJF variance, using the Nino3 index in the Pacific. The numbers are provided in Table 1 below.

| | PI | MHREF | NODUST | Albani0k | Albani6k |
|---|---|---|---|---|---|
| Magnitude of seasonal cycle | 3.0 | 1.86 | 1.83 | 1.95 | 1.86 |
| Variance | 1.07 | 0.85 | 0.75 | 0.73 | 0.86 |

Table 1. Estimates of the magnitude of the seasonal cycle and the DJF interannual variance in the Ninõ3 box for the different simulations with the IPSLCM6-LR model. The magnitude of the seasonal cycle is estimated by the difference between the maximum and minimum monthly values of the

annual mean seasonal cycle estimated over the whole duration of the simulations (500 years for PI and MHRF, 290 years for Albani6k and 200 years for NODUS and Albani0k).

The estimates from Table 1 show that between mid-Holocene and pre-industrial climates there is both a roughly 35 to 39% reduction of the magnitude of the seasonal cycle and a reduction of 20 to 30% of the ENSO variability amongst the mid-Holocene simulations. This is consistent with the estimates by Brown et al., Climate of the past, 2020. These numbers also show that we only have tiny differences between the mid-Holocene simulations. Going further would require a much more extensive statistical analysis. The results from Table 1 indicate that there is no direct rationale with dust reduction/increase from a reference simulation, even though the NODUST simulation provides lower (possibly non-significant) values. We also think that the length of NODUST and Albani0k (200 years) is sufficient for the detailed analyses of the forcing and energetic we provide here, but it might not be sufficient to properly analyse ENSO variability and detect small changes as the ones emerging from the crude estimates provided in Table 1 (see for example Stevenson et al. (2010). ENSO Model Validation Using Wavelet Probability Analysis, *Journal of Climate*, *23*(20), 5540-554). Indeed, we also know from our own experience with long simulations with the IPSL model that the model produces a large diversity in ENSO variability with several decades or centuries with low or high variability (see Braconnot et al. GRL, 2019).

We would prefer not to add a discussion on ENSO to this already extensive study, but we will mention when discussing the changes in the Pacific (Figure 6) that the differences are small over the region and that mid-Holocene simulations show no major differences in variability reduction.

Compared to Pausata et al 2016, these simulations do not have the huge increase in monsoon found in the Ec-Earth simulations and that mostly results from the huge changes in vegetation imposed over the Sahel-Sahara regions that seem to trigger intense teleconnexions. Hence, we cannot draw strong conclusions on ENSO and dust from our sensitivity experiments.

Response to the other comments

*L.13 mid-Holocene: the expression of mid-Holocene is not consistent in the manuscript (mid Holocene, midHolocene), standardize it.*
We agree that this is a problem. We oscillated between mid-Holocene, midHolocene which correspond to the standard name of PMIP simulation. We also agree that for the manuscript we need to simplify a little bit. So, the rationale is now that mid-Holocene refers to the period and it is abbreviated as MH.

*L. 29: Introduction discusses only the issue of dust in mid-Holocene but a broader context would attract wider readers. Discuss the role of dust on climate of the past and future and why mid-Holocene are important among the different times.*
We are adding a discussion of this aspect in the introduction. Referring to the different factors affecting dust emission e.g. wind gusts, soil moisture, vegetation cover, availability of erodible material, snow/ice cover, all of which may vary in the future due to both climate change and land use and land cover changes (Mahowald, 2007 ; Stanelle et al. 2014; Evan et al. 2016; Ginoux et al. 2012; Bullard et al. 2016), and slightly reinforce the discussion for the mid-Holocene.

Bullard, J. E., Baddock, M., Bradwell, T., Crusius, J., Darlington, E., Gaiero, D., Gassó, S., Gisladottir, G., Hodgkins, R., McCulloch, R., McKenna-Neuman, C., Mockford, T., Stewart, H. and Thorsteinsson, T.: High-latitude dust in the Earth system, Rev. Geophys., 54(2), 447–485, https://doi.org/10.1002/2016RG000518, 2016.

Evan, A. T., Flamant, C., Gaetani, M. and Guichard, F.: The past, present and future of African dust, Nature, 531(7595), 493–495, https://doi.org/10.1038/nature17149, 2016.

Ginoux, P., Prospero, J. M., Gill, T. E., Hsu, N. C. and Zhao, M.: Global-scale attribution of anthropogenic and natural dust sources and their emission rates based on MODIS Deep Blue aerosol products, Rev. Geophys., 50(3), https://doi.org/10.1029/2012RG000388, 2012.

Mahowald, N. M.: Anthropocene changes in desert area: Sensitivity to climate model predictions, Geophys. Res. Lett., 34(18), https://doi.org/10.1029/2007GL030472, 2007.

Stanelle, T., Bey, I., Raddatz, T., Reick, C. and Tegen, I.: Anthropogenically induced changes in twentieth century mineral dust burden and the associated impact on radiative forcing, J. Geophys. Res. Atmospheres, 119(23), 13,526-13,546, https://doi.org/10.1002/2014JD022062, 2014.

*L.71 mid mid-Holocene –> mid-Holocene*
*See above*

*L. 73 Can you explain briefly what INCA-IPSL model is?*
INCA is the atmospheric aerosol and chemistry model developed at IPSL. We will add a few words explaining it in the text.

*L. 99 Two expressions are mixed in each experiment in the manuscript, making it difficult*
*to understand: MHREF, NODUST, Albani0k, Albani6k and f1, f2 : : :. Use one type.*
*I prefer MHREF : : :.*
Sorry about this, it took some time to adjust and make sure that we do not lose the reference to the mid-Holocene member posted on ESGF, for which f refer to the fact that the simulation differ by their forcing field and the number to the field itself, an easy name to characterize dust files, and a short name that can be easily used to characterized the simulation and can be used in figure titles. This is now clear that we call the simulation MHREF, NODUST, Albani0k and Albani6ka, and that the other names will only be provided in the table and in the text when we explain these simulations.

*L. 115 a slight reduction — I have an impression nearly 30 % of reduction is more than*
*"a slight reduction"*
Agreed. Precision will be added on this statement.

*L. 253 DJF: It is not clear that the calendar adjustment (Otto-Bliesner et al. 2017 GMD)*
*is applied for the seasonal analyses throughout the study.*
The calendar adjustment is not considered here. This is also a reason why (and figure 7 confirms the choice) we decide to only use JA and JF. The calendar effect is small for mid-Holocene, and we know from experience and previous PMIP multi-model publications, that JA is well suited for the analyses of the summer Northern hemisphere monsoon season in India and Africa, because it captures the peak monsoon that arise either in July or August depending of the period or monsoon region considered. The other reason is that the core of the paper is a detailed analysis of the energetics, using different approaches for which the most important point is the energy consistency between the climate variables. A posteriori calendar correction would introduce errors in the energetic as large or larger than the small error we have for seasonal averages (I.E considering months that have similar "phasing" with respect to the insolation forcing whatever the paleo-period of interest). The impact of dust we consider here is larger than the calendar effect for mid-Holocene. The most important point in these comparisons is the centennial variability which is larger than the calendar effect, except it is not a systematic effect.

In the revision the calendar issue will be raised at the beginning of the paper and we will redo figure 4, 5 and 6 with JA and JF, so that all the analyses and figures are consistent. It was not done because these figures were initially drawn using a standard atlas procedure, for which seasonal averages were pre-defined. For these figures using DJF or JJA doesn't change the results and the discussion on the figure, just the magnitude of some of the maxima because JF and JA better correspond to peak seasons.

*L. 259 Several important differences appear – Explain what the important differences how does they mean.*
The vocabulary will be refined and the differences better quantified.

*L 261: The summer monsoon precip: : : – Judged from Fig 5, the precipitation over Sahara seems greatly improved than previous modelling studies. Can you explain how much improved from the previous models and why? I could not distinguish pronounced improvement of the IPSL model at Fig. 8 of Brierley et al. CP 2020. Please discuss on it.*

The representation of the African rainbelt has indeed been improved in this version of the IPSL model. This is true for the modern climate and for the mid-Holocene. For the modern climate it doesn't mean that there is no bias left. It means that the location of the rainbelt is in better agreement with observations, and that there is still a dry bias, but that is much less pronounced than before in the Sahel (see Boucher et al., James, 2020). For the mid-Holocene the improvement is that the precipitation can penetrate further inland. However, the isolines also indicate that most of it are still with very small amounts. In addition, when comparing the two simulations on the large box considered either in Brierley et al., Cliamte of the Past 2020 (figure 8) or in figure 14 there are several places where the amount is slightly smaller in CMIP6 than in CMIP5. It explains why the large box doesn't reflect well this improvement. It is also clear that whatever we do for these comparisons, we are far from the large values suggested by the paleoclimate reconstructions.
In complement of this response, we provide a version of PMIP spaghetti diagram showing the model-data comparison over west Africa. This figure was part of a poster presentation of the IPSL results in France, 2 years ago. We included also the modern location of the rainbelt. It indeed shows the improvement of the CMIP6 version compared to the CMIP5 one and the fact that all the f members are different from the CMIP5 simulations in terms of magnitude and northward extent. We will not include this figure in the manuscript because we do not have equivalent figures for the other regions. In addition, the simple averages we made over the "Sahel" reflect the differences we have here

between the two model versions and the sensitivity experiments.

[Figure]

This comment and the comment on the conclusions however suggest that we move part of the discussion on model-data comparison at the end of the section on the comparison between CMIP5 and CMIP6 simulations focusing only on the differences between the two model versions in section 3.1 , and then have the second part of the discussion dedicated to the effect of the different dust sensitivity experiments when discussing the differences between the sensitivity experiments in section 3.2. We will revisit the text on the model improvement in the Sahel region, referring to relevant publications that are now available.

Note that the model-data comparisons are not the core of this study. We included it just to say that there is nothing to say on model-data compared to what we already know from other studies and that dust doesn't provide the right order of magnitude to improve the model-data comparisons. Going further in the analyses would be interesting, but is clearly a different paper. It would bring too much distraction from the core energetic analyses and detailed analyses of the dust impact. Therefore, we will change the way we discuss the model -data comparison, but will not put a larger focus on it.

*L 261: data reconstruction – give references*

Done

*L 342: JF – why do you choose January-February instead of DJF? The same for JA.*

See comment on the calendar issue. This is done to have the peak monsoon seasons that is valid both for modern and mid-Holocene climate. As proposed above we will redo figure 4 to 6 using JA and JF and explaining right from the beginning that we do not consider the calendar correction.

*L 597: Bartlein et al. 2011: Kaufman et al. 2020 Scientific Data,*
*https://doi.org/10.1038/s41597-020-0445-3 provides narrower constraints than*
*Bartlein.*
We agree with this. The point is that using a new dataset would bring us in a new territory for which we do not yet have the expertise on the differences between the datasets and in the way to consider uncertainties, and lots of other details that are out of the scope of this study. We do not have the resources to do more for this paper. We propose to keep what we have an enlarge a little bit the discussion on the model side, and better reference the data, as mentioned above. The model-data is here to show that the signal we are looking at with dust is not of the order of magnitude that would be needed to improve model-data agreement. A more complete model-data comparison using different datasets and providing guidance on the reason of model-data mismatch is out of the scope of this paper. The basic model-data comparisons for CMIP5 and CMIP6 PMIP simulations can be found in Brierley et al. 2020. We will reinforce the link with this paper in the manuscript.

*L 598 paleoclimate reconstruction from pollen and macro fossil data (Fig.13). – Figure*
*13 does not explain paleoclimate reconstruction.*
Corrected.

*Fig. 14 is only discussed in the Discussion section. But I think the Discussion and*
*Conclusion should be a summary of the entire paper. I suggest that Discussion and*
*conclusion to start from Line 608. And add one section beforehand for from Line 585*
*to Line 607.*

We propose to revise the manuscript and reorder the figures to have the model-data comparison discussed in section 3.1 and 3.2 respectively for the comparison between the CMIP5 and CMIP6 version of the IPSL model and the impact of the different dust forcing.

*Line 615: mid-Holocene dust reduction –> mid-Holoicene*
Done

*Table 1: If you call experiments by dust file, reform the table. For clarity, it is recommended*
*to put MHREF, NODUST, etc. in the first column (simulation name), and the*
*ripf code and f1 : : : can be in the last column. The same for Table 2.*
See comment above, this will be done.

*Table 2: -0.-06 –> -0.06 ?*
Typo corrected

*Figure 1: labels and legends are too small to read. Figure 1 (b): There exist clear*
*negative trend of global mean temperature. This means that the experiments are not*
*equilibrium. In such cases, I think it is important to align the time periods of the original*
*experiment and the analysis of the branched group of experiments. In addition to the*
*lack of equilibrium, there is another reason why you should take the same period. From*
*(h), the 100-year variability of the AMOC is so large that if you take different periods,*
*you may see differences in the AMOC state due to different phases.*

We are not sure to understand all the points in the comment. Each simulation starts from the right initial state. This has been properly done, and it corresponds to what is indicated in Table 1, and the way the simulations have been done. The effect of trend in temperature is mainly due to the fact that indeed there is a trend for the adjustment period, which is the red curve. The trend is very small afterwards.

We will refine the arguments on variability for the choice of the common reference period, because the comment let us think it is not clear enough. It is important, because it also explains why we used different 150-year periods in the green simulation to assess variability and not only an estimate of variability within the 150-year period. This is a key point in the analyses of this paper on which we have spent quite a lot of time to make sure we provide clean analyses and not a result that could only be due to variability and not to dust forcing.

*In Figure 1 some experiments are only shown for 200 years while in table 1 all experiments are shown for more than 290 years.*
Sorry there was a typo in the table that has led to an error in the text and that should have been corrected before the submission of the manuscript. The figure is correct. Two simulations are indeed 200-year long and not 300 as written. Only Albani6k should have been 300 years. The additional 100 years compared to the other sensitivity experiments are there to check that for these tests 200 year are sufficient, which is the case. We chose to have a longer simulation for Albani6k, because this is also the simulation with the mid-Holocene dust compared to the other two simulations. The reason why it is finally 290 and not 300 are related to computing constraints as explained in the manuscript.

Fig. 2: Add proper reference at REF.*Figure 3: Make the figures bigger and fill in the gaps to make them easier to read*.
The figure has been revisited and the references added in the legend as requested.

*Figure 6: How about a narrower range of temperatures? The colours in the diagram could be clearer and easier to read.*
The objective was to be able to directly compare the effect of dust with that of insolation. But it is clear that we adjusted the color and contours for figure 10 and 12. This will be revisited, trying to maintain for these surface fields a direct comparison between the role of the insolation forcing and the impact of the dust forcing.

*Figure 7: c, d, g, h – the highest and the lowest colors are difficult to distinguish.*
This has been done on purpose, because it refers to the same season. There is no reason to distinguish December from January too much, at least for our objectives. This will be kept as it is. Several tests have already been made for this figure and it is the best solution we had.

*Figure 9: c panels are almost the same with a. There are quite a lot of figures in this study. The comparison of a and c could be at supporting figures?*
Thank you for helping us figuring out how we can reduce the number of figures while making it easy to convince the reader that the same applies between the two estimates.
We would like to avoid having supporting figures that are rarely looked at for a Climate of the Past manuscript where it is permitted to have enough figures to convey the messages. Instead, we propose to include an additional figure in the Appendix, showing only NODUST vs MHREF for JF and JA comparing the Taylor et al. method with irf estimates.

Then we will suppress the last row figure 9a and 9b and provide a new version with JF and JA all together on the same page.

*Figure 10: provide unit*.
Thank you for noticing it.

*Figure 11: This is very busy figure. In order to get the point across, you need to make it easier to read. At least, I cannot read numbers at blue lines. Some of the numbers on color bars are half hidden. Give a, b, c : : : for each panel.*

We will keep this figure busy, because it is a way to show connections between the different factors, and the fact that the effect of dust is the combination of lots of factors that cannot be easily disentangled. We have to fix a graphic bug for the blue line (the dotted blue lines do not appear as they should when printed) and indeed larger numbers will help. Letters will be added to the different panels that were done independently and assembled for the figure.

Figure 12: As in Figure 10 – > As in Figure 11
The figure will be revised

*Figure 13: Should yellow color start from 0.25 Sv? Mask non-oceanic grids to black or gray. Southern boundary could be 35S? Uniformly dotting the stippling entire panel improves visibility.*
The figure suffers in the transfer to pdf format. Dots are plotted at each model grid point in our case, because pattern is not working with the version of pyferret we are using to draw this figure. All the other changes will be included in this figure.

*L. 690: According to the data policy of climate of the past, the authors are requested to upload the data and codes for the analyses to a reliable data repository (or as supplement materials) so that anyone can reproduce the data in figures and tables. The authors are also requested to provide DOIs for the data on the ESGF.*

The DOI will be provided.